# Beyond-Expert Performance with Limited Demonstrations: Efficient Imitation Learning with Double Exploration

**Heyang Zhao**[1*], **Xingrui Yu**[23*], **David M. Bossens**[23*], **Ivor W. Tsang**[23†], **Quanquan Gu**[1†]
[1]Department of Computer Science, University of California, Los Angeles
[2]IHPC, Agency for Science, Technology and Research, Singapore
[3]CFAR, Agency for Science, Technology and Research, Singapore
`{hyzhao,qgu}@cs.ucla.edu`
`{yu_xingrui,david_bossens,ivor_tsang}@cfar.a-star.edu.sg`

## Abstract

Imitation learning is a central problem in reinforcement learning where the goal is to learn a policy that mimics the expert's behavior. In practice, it is often challenging to learn the expert policy from a limited number of demonstrations accurately due to the complexity of the state space. Moreover, it is essential to explore the environment and collect data to achieve beyond-expert performance. To overcome these challenges, we propose a novel imitation learning algorithm called Imitation Learning with Double Exploration (ILDE), which implements exploration in two aspects: (1) optimistic policy optimization via an exploration bonus that rewards state-action pairs with high uncertainty to potentially improve the convergence to the expert policy, and (2) curiosity-driven exploration of the states that deviate from the demonstration trajectories to potentially yield beyond-expert performance. Empirically, we demonstrate that ILDE outperforms the state-of-the-art imitation learning algorithms in terms of sample efficiency and achieves beyond-expert performance on Atari and MuJoCo tasks with fewer demonstrations than in previous work. We also provide a theoretical justification of ILDE as an uncertainty-regularized policy optimization method with optimistic exploration, leading to a regret growing sublinearly in the number of episodes.

## 1 Introduction

Imitation learning (IL) is an important subfield of reinforcement learning (RL), in which ground truth rewards are not available and the goal is to learn a policy based on expert demonstrations. Often such demonstrations are limited, making it challenging to achieve expert-level performance. In practice, imitation learning is widely used in various applications such as autonomous vehicle navigation (Codevilla et al., 2018), robotic control (Finn et al., 2016; Zhang et al., 2018) and surgical assistance (Osa et al., 2014; Li et al., 2022).

As one of the simplest approaches of imitation learning, behavior cloning (Pomerleau, 1988) learns a policy directly from the state-action pairs in the demonstration dataset. More recent approaches to imitation learning, such as generative adversarial imitation learning (GAIL) (Ho & Ermon, 2016), use a discriminator to guide policy learning rather than directly learning a reward function based on demonstration trajectories. Due to matching occupancy based on a limited set of trajectories, it is still difficult for such algorithms to achieve better-than-expert performance and solve high-dimensional problems with limited demonstration data. Moreover, these techniques are typically unstable since the reward function is continually changing as the reward function is updated during the RL training. Alternative approaches, based on the curiosity (Yu et al., 2020; Pathak et al., 2017; Burda et al., 2018), a form of self-supervised exploration, encourage the agent to explore transitions that are distinct from the expert – thereby potentially yielding beyond-expert performance. Another benefit of the approach is related to the GIRIL algorithm specifically; that is, it uses a pre-trained auto-encoder, which makes the technique more stable (Yu et al., 2020).

---

*Equal technical contribution. †Corresponding author.

As illustrated in Table 3 in Appendix A, this work seeks to formulate a sample-efficient imitation learning algorithm that combines the best of both worlds. That is, we integrate the stability and self-supervised exploration properties of GIRIL into discriminator-based imitation learning approaches. Additionally improving the sample-efficiency with an exploration-bonus, we formulate the framework of Imitation Learning with Double Exploration (ILDE).

In detail, ILDE learns a policy based on an uncertainty-regularized discrepancy. The uncertainty-regularized discrepancy combines the distance to the expert policy, the cumulative uncertainty of the policy during exploration, and the cumulative uncertainty of the policy with respect to the demonstration dataset. To optimize the uncertainty-regularized discrepancy, ILDE utilizes a GAIL-based imitation reward while integrating two distinct forms of exploration. Firstly, through optimistic policy optimization, augmented by an exploration bonus, ILDE incentivizes the exploration of state-action pairs characterized by high uncertainty, thereby facilitating the training of the discriminator network, which functions as the reward model. Secondly, leveraging curiosity-driven exploration, ILDE targets transitions deviating from demonstration trajectories, paving the way for nuanced policy improvements.

The key contributions of this paper are summarized as follows:

- In Section 3, we theoretically formulate the problem of solving the aforementioned policy which minimizes the uncertainty-regularized discrepancy between the learner trajectories and demonstration trajectories.

- In Section 4, we present our proposed algorithm, ILDE with natural policy gradient (ILDE-NPG). We are then able to demonstrate the regret guarantee of ILDE-NPG in Theorem 4.7, which is the first theoretical guarantee for imitation learning in MDPs with nonlinear function approximation.

- Although ILDE-NPG is computationally expensive, we offer a more flexible version for the practical implementation of ILDE in Section 5, focusing on two key exploration strategies: (1) encouraging the learner to explore state-action pairs that are 'far' from the current rollout dataset, and (2) rewarding benign deviations from the demonstration dataset using curiosity-based intrinsic rewards.

- In Section 6, we present experimental results for a practical implementation of ILDE. Our findings demonstrate that ILDE surpasses existing baselines in terms of both average return and sample complexity. This underscores the practical advantages of our proposed objective for imitation learning.

## 2 RELATED WORK

**Imitation learning with limited demonstrations for beyond-expert performance.** Learning from limited demonstrations in a high-dimensional environment is challenging, and the effectiveness of imitation learning methods is hampered (Li et al., 2023). Inverse reinforcement learning (IRL) methods (Ziebart et al., 2008; Ziebart, Brian D. and Bagnell, J. Andrew and Dey, Anind K., 2010; Boularias et al., 2011) seek a reward function that best explains the demonstration data, which makes it hard for them to achieve better-than-expert performance when the data are extremely limited. GAIL achieves impressive performance in low-dimensional environments via adversarial learning-based distribution matching. Variational adversarial imitation learning (VAIL) improves GAIL by compressing the information flow with a variational discriminator bottleneck (Peng et al., 2019). Recent adversarial imitation learning methods include IQ-Learn (Garg et al., 2021), which directly infers the Q-value (bypassing the reward function), and HyPE (Ren et al., 2024), which trains the policy on a mixture of expert and on-policy data.

Unfortunately, GAIL and VAIL do not scale well in high-dimensional environments (Brown et al., 2019), and still require many episodes of demonstrations. Recent work has focused on reducing the number of required demonstration episodes to between 5 and 20 episodes using techniques such as MCTS-based RL (Yin et al., 2022), patch rewards (Liu et al., 2023a), and applying demonstration augmentation and a prior policy baseline Li et al. (2022). Techniques based on curiosity-driven exploration, such as CDIL (Pathak et al., 2017) and GIRIL (Yu et al., 2020), potentially provide a more scalable approach to imitation learning. In particular, Yu et al. (2020) propose the GIRIL algorithm within a setting with only a so-called one-life demonstration, which is only a partial episode of an Atari game. The system demonstrates favorable performance on Atari games when compared to GAIL, VAIL, and CDIL. Such techniques with intrinsic motivation have the advantage of allowing beyond-expert performance but the disadvantage of potentially optimizing an objective that is not

implied by the demonstrations. Our proposed ILDE system seeks to use curiosity-driven exploration as one of two sources of exploration to supplement a traditional imitation learning objective. This is based on the fact that the curiosity reward has high empirical success and enables agents to learn non-trivial skills by exploring hard-to-visit states (Rajaraman et al., 2020).

Beyond reducing the sheer amount of demonstration data, the problem of varying optimality scores and the reliability of demonstrations has also garnered more attention recently. Confidence-Aware Imitation Learning (CAIL) extends adversarial inverse reinforcement learning with a bi-level optimization technique to concurrently learn the policy along with confidence scores reflecting the optimality of the demonstrations (Zhang et al., 2021). The setting of imbalanced demonstrations has been studied from a semi-supervised learning perspective (Fu et al., 2023) as well as within multi-modal imitation learning (Gu & Zhu, 2024). We focus on an alternative approach to the problem of demonstration reliability which combines curiosity with a traditional imitation reward.

**Reinforcement learning from demonstrations (RLfD).** An alternative framework, related to imitation learning, is reinforcement learning from demonstrations (RLfD) (Hester et al.; Christiano et al., 2017; Zhu et al., 2022). While potentially offering accelerated policy learning, such techniques require ground-truth rewards, which are not available in pure imitation learning settings.

**Exploration bonuses and optimistic RL.** Optimism is a long-standing principle in RL, where it typically refers to overestimating the value of a particular state or state-action pair in order to make the state or state-action pair known more rapidly through increased visitations – essentially, this implies that unknown state-action pairs are given a large exploration bonus while known state-action pairs are given little to no exploration bonus. Analogous to upper confidence bound (UCB) bandit algorithms (Auer et al., 2002), various techniques have been implemented in RL which use confidence bounds to upper bound the value in a probabilistic sense. Such techniques traditionally use concentration inequalities based on visitation counts for each state-action pair, which gives such techniques a firm statistical grounding but which limits them to discrete state-action spaces (Shani et al., 2020; Fruit et al., 2018; Jaksch et al., 2010).

Recent techniques have aimed at making exploration bonuses more scalable. E3B provides an approximate approach to visitation counts by maintaining a covariance matrix which implicitly captures the relations between visited states (Henaff et al., 2023). State Entropy directly aims to maximze the entropy over the states, rewarding states in a batch according to the log of their distance to its $k$-nearest neighbor state (Seo et al., 2021).

**Theoretical guarantees for imitation learning.** There is a large body of literature on imitation learning for tabular MDPs (Cai et al., 2019; Chen et al., 2019; Zhang et al., 2020; Xu et al., 2020; Shani et al., 2022; Chang et al., 2021). Subsequently, Liu et al. (2021) studied imitation learning for linear kernel MDPs and provided an $\widetilde{O}(\sqrt{H^4 d^3 T})$ regret bound, where $d$ is the dimension of the feature space, $H$ is the horizon of the MDPs, $T$ is the number of episodes. Viano et al. (2022) considered linear MDPs and proposed PPIL, which achieved a sample complexity of $O(d^2/(1-\gamma)^9 \epsilon^5)$ in discounted linear MDPs. Our work considers MDPs with general function approximation, utilizing the generalized Eluder dimension (Agarwal et al., 2023; Zhao et al., 2023) to characterize the complexity of the function class.

## 3 PRELIMINARIES

We consider an episodic MDP $(\mathcal{S}, \mathcal{A}, H, \mathbb{P}, r)$, where $\mathcal{S}$ and $\mathcal{A}$ are the state and action spaces, respectively, $H$ is the length of each episode, $\mathbb{P}_h$ is the Markov transition kernel of the $h$-th step of each episode for any $h \in [H]$, and $r : S \times A \to [-1, 1]$ is the reward function. We assume without loss of generality that the reward function $r$ is deterministic.

While performing imitation learning in episodic MDPs, the agent interacts with the environment as follows. At the beginning of each episode $t \in [T]$, the agent chooses a policy $\pi^t = \{\pi_h^t\}_{h \in [H]} \in \Delta(\mathcal{A}|\mathcal{S}, H)$. Then the agent takes an action $a_h^t \sim \pi_h^t(\cdot|s_h^t)$ at the $h$-th step of the $t$-th episode and transits to the next state $s_{h+1}^t \sim \mathbb{P}_h(\cdot|s_h^t, a_h^t)$. The agent does not receive the true reward $r^*(s_h^t, a_h^t)$ but instead it receives a surrogate reward $r(s_h^t, a_h^t)$. The episode terminates when the agent reaches the state $s_{H+1}^t$. Without loss of generality, we assume that the initial state $s_1 = x$ is fixed across different episodes. We remark that our algorithms and corresponding analyses readily generalize to the setting where the initial state $s_1$ is sampled from a fixed distribution.

We now define the value function in episodic MDPs. For any policy $\pi = \{\pi_h\}_{h \in [H]}$ and reward function $r : \mathcal{S} \times \mathcal{A} \to [-1, 1]$, the state value function $V$ and action-value function $Q$ are defined for any $(s, a, h) \in \mathcal{S} \times \mathcal{A} \times [H]$ as follows,

$$V_{h,\pi}^r(s) = \mathbb{E}_{\pi_h}\Big[\sum_{i=h}^H r(s_i, a_i) \mid s_h = s\Big], \quad Q_{h,\pi}^r(s, a) = \mathbb{E}_{\pi_h}\Big[\sum_{i=h}^H r(s_i, a_i) \mid s_h = s, a_h = a\Big], \tag{3.1}$$

where the expectation $\mathbb{E}_\pi[\cdot]$ is taken with respect to the action $a_i \sim \pi_i(\cdot | s_i)$ and the state $s_{i+1} \sim \mathbb{P}_i(\cdot | s_i, a_i)$ for any $i \in \{h, h+1, \ldots, H\}$. With slight abuse of notation, we also denote by $\mathbb{P}_h$ the operator form of the transition kernel such that $(\mathbb{P}_h f)(s, a) = \mathbb{E}_{s' \sim \mathbb{P}_h(\cdot | s, a)}[f(s')]$ for any $f : S \to \mathbb{R}$. By the definitions of the value functions in (3.1), for any $(s, a, h) \in S \times A \times [H]$, any policy $\pi$, and any reward function $r$, we have

$$V_{h,\pi}^r(s) = \langle Q_{h,\pi}^r(s, \cdot), \pi_h(\cdot, s)\rangle_A, \quad Q_{h,\pi}^r(s, a) = r_h(s, a) + \mathbb{P}_h V_{h+1,\pi}^r(s, a), \quad V_{H+1,\pi}^r(s) = 0, \tag{3.2}$$

where $\langle \cdot, \cdot \rangle_A$ denotes the inner product over the action space $A$. We further define the expected cumulative reward as follows,

$$J(\pi, r) = V_{1,\pi}^r(x).$$

We assume that there is an unknown expert policy $\pi^E = \{\pi_h^E\}_{h \in [H]} \in \Delta(\mathcal{A} | \mathcal{S}, H)$ that achieves a high expected cumulative reward $J(\pi^E, r^*)$ under the unknown underlying reward function $r^*$. Given $n$ demonstration trajectories $\tau^E = \{(s_i^{(j)}, a_i^{(j)})\}_{i=1}^H$ for $j \in [n]$, the goal of imitation learning is to learn a policy $\pi$ that achieves a potentially high expected cumulative reward $J(\pi, r^*)$ under the unknown reward function $r^*$ based on the expert demonstration. As introduced in Chen et al. (2019), we characterize the discrepancy between the expert policy $\pi^E$ and the learner policy $\pi$ by the following Integral Probability Metric (IPM) over the stationary distributions of the MDP.

**Definition 3.1** (Definition 2, Chen et al. 2019). Let $\mathcal{R}$ denote a class of symmetric reward functions $r : \mathcal{S} \times \mathcal{A} \to [-1, 1]$, i.e., if $r \in \mathcal{R}$, then $-r \in \mathcal{R}$. Given two policies $\pi, \pi' \in \Delta(\mathcal{A} | \mathcal{S}, H)$, the $\mathcal{R}$-distance is defined as

$$d_{\mathcal{R}}(\pi, \pi') = \sup_{r \in \mathcal{R}} \big| J(\pi^E, r) - J(\pi, r)\big|.$$

**Remark 3.2.** The IPM distance is a versatile tool for evaluating GAN models. In the context of imitation learning, we can choose different classes of reward functions $\mathcal{R}$ to measure the discrepancy between the expert policy and the learner policy. For instance, we can choose $\mathcal{R}$ to be the class of symmetric reward functions that are 1-Lipschitz continuous with respect to the state-action pair $(s, a)$, which corresponds to the Wasserstein distance. Or we can choose $\mathcal{R}$ to be the unit ball in an RKHS, which yields kernel maximum mean discrepancy (MMD).

As proposed in Yu et al. (2020), to encourage the agent to explore the environment and learn a beyond-expert policy, it is essential to incorporate *uncertainty-driven* (intrinsic-reward-driven) exploration into the imitation learning framework. In contrast to GIRIL (Yu et al., 2020) which purely relies on the intrinsic reward to explore transitions that are distinct from the expert demonstration, we consider learning a policy which mimics the expert policy subject to an intrinsic reward regularisation term. To this end, we defined the following objective function:

$$\min_{\pi \in \Delta(\mathcal{S} | \mathcal{A}, H)} \max_{r \in \mathcal{R}} \big(J(\pi^E, r) - J(\pi, r) - \lambda \cdot \text{Int}(\pi; \tau^E)\big), \tag{3.3}$$

where $\text{Int}(\pi; \tau^E)$ is the expected cumulative intrinsic reward of the policy $\pi$ under the expert demonstration $\tau^E$. More specifically,

$$\text{Int}(\pi; \tau^E) := \mathbb{E}_{\pi, \widehat{s}_{h+1} \sim \widehat{\mathbb{P}}(\cdot | s_h, a_h)}\Big[\sum_{h=1}^H \mathcal{L}(\widehat{s}_{h+1}, s_{h+1})\Big],$$

where $\widehat{\mathbb{P}}$ is the estimated transition probability that is estimated as follows:

$$\widehat{\mathbb{P}} := \operatorname*{argmin}_{\mathbb{P} \in \mathcal{P}^{\text{model}}} \sum_{j=1}^n \sum_{i=1}^H \mathbb{E}_{\widehat{s}_{i+1} \sim \mathbb{P}(\cdot | s_i^{(j)}, a_i^{(j)})}\Big[\mathcal{L}(\widehat{s}_{i+1}, s_{i+1}^{(j)})\Big]$$

is a trained transition model over the demonstration trajectories $\tau^E$, and $\mathcal{L}$ is a distance metric (e.g. $\mathcal{L}(\widehat{s}_{i+1}, s_{i+1}^{(j)}) = ||\widehat{s}_{i+1} - s_{i+1}^{(j)}||_2^2$). (Empirically, it is the trained VAE model which samples the next state as proposed in Yu et al. (2020) and Pathak et al. (2017).) For simplicity, we denote the intrinsic reward as the following form

$$\mathcal{L}_{\tau^E, h}(s, a) = \mathbb{E}_{\widehat{s}' \sim \widehat{\mathbb{P}}_h(\cdot|s,a), s' \sim \mathbb{P}_h(\cdot|s,a)}\left[\mathcal{L}(\widehat{s}', s')\right]. \tag{3.4}$$

As a shorthand, we define the **uncertainty-regularized loss function** for a policy $\pi$ and a reward function $r$ as follows,

$$\ell(\pi, r) = J(\pi^E, r) - J(\pi, r) - \lambda \cdot \text{Int}(\pi; \tau^E). \tag{3.5}$$

**Imitation Learning.** As shown in (3.3) and (3.5), our goal is to learn an optimal policy $\pi^*$ which minimizes the following worst-case loss:

$$\pi^* = \underset{\pi \in \Delta(\mathcal{A}|\mathcal{S}, H)}{\text{argmin}} \max_{r \in \mathcal{R}} \ell(\pi, r). \tag{3.6}$$

**Conjecture.** To achieve the optimal policy as defined in (3.6), we hypothesize that a more principled exploration technique is needed to adapt to the dynamic rewards during the training process. Additionally, we conjecture that such uncertainty regularization will help the learner attain performance beyond the expert level.

In the previous theoretical formulation of online imitation learning (Chen et al., 2019; Liu et al., 2021), the objective is to learn a policy which is close to the expert policy with respect to the $\mathcal{R}$-distance. Thus, the optimal policy under their formulation is $\pi^E$ and the loss at the saddle point is zero. By contrast, our optimal policy $\pi^*$ is non-trivially different from the expert policy $\pi^E$.

In this work, we aim to find $\pi^*$ by proposing a novel theory-guided policy optimization algorithm with uncertainty-aware exploration.

Within the online learning setting, we define the uncertainty regularized regret as follows:

$$\text{Regret}(T) = \max_{r \in R} \sum_{t=1}^{T} \ell(\pi^t, r) - \ell(\pi^*, r).$$

To achieve the objective (3.3) in the online learning regime, we aim to design an imitation learning algorithm to achieve a $\text{Regret}(T)$ growing sublinear as a function of $T$. Then we can show that $\mathbb{E}[\ell(\pi_{\text{out}}, r)]$ converges to $\ell(\pi^*, r)$ as $T \to \infty$ for any $r \in \mathbb{R}$, where $\pi_{\text{out}}$ refers to the hybrid policy sampled uniformly from $\{\pi^t\}_{t \in [T]}$.

# 4 IMITATION LEARNING WITH DOUBLE EXPLORATION (ILDE)

We now turn to introducing the ILDE framework theoretically before its implementation in Section 5. We analyze ILDE using mirror descent as the policy optimization technique which is more amenable to theoretical analysis than scalable techniques such as PPO. The theoretical analysis considers a general data collection subroutine of which on-policy data collection used in the practical implementation is a special case. The theoretical analysis further considers an IPM-based GAN and a bonus based on state-action uncertainty as proposed by (Agarwal et al., 2023). In practice, we apply GAIL and state entropy (Seo et al., 2021). Since PPO already applies action entropy in the policy optimization, together the state entropy and action entropy help to explore unvisited state action pairs in a similar manner.

## 4.1 ALGORITHM

This subsection instantiates a theoretical version of ILDE, where we apply Optimistic Natural Policy Gradient (Liu et al., 2023b) as a policy optimization module. Since Algorithm 1 is a phasic policy optimization algorithm, the number of episodes where the learner interacts with the environment is $T = K \cdot N/m$.

**Imitation reward Module.** In Line 4 of Algorithm 1, the learner samples a data set with $N$ trajectories. With a carefully chosen $N$, the expected uncertainty of each state-action pair $(s, a) \in$

---

**Algorithm 1** ILDE with Natural Policy Gradient

---

1: **input**: number of iterations $K$, period of collecting fresh data $m$, batch size $N$, learning rate $\eta$, demonstration trajectories $\tau^E$, reward function $r^1$, loss function $\mathcal{L}_{\tau^E}$.
2: **initialize:** for all $(h, s) \in [H] \times \mathcal{S}$ set $\pi_h^1(\cdot \mid s) = \text{Uniform}(\mathcal{A})$
3: **for** $k = 1, \ldots, K$ **do**
4:     **if** $k \bmod m = 1$ **then**
5:         $\mathcal{D}^k \leftarrow \{N \text{ fresh trajectories} \overset{\text{i.i.d.}}{\sim} \pi^{k'}\}$, where $k'$ is chosen uniformly at random from $\{\max(1, k - m + 1), \ldots, k - 1, k\}$.
6:     **else**
7:         $\mathcal{D}^k \leftarrow \mathcal{D}^{k-1}$.
8:         $r^k \leftarrow r^{k-1}$.
9:     **end if**
10:    Update $r^k$ by projected gradient descent with estimated $-L(\pi^k, r^k)$.
11:    $\{Q_h^k\}_{h \in [H]} \leftarrow \text{OPE}(\pi^k, \mathcal{D}^k, r^k, \mathcal{L}_{\tau^E})$.
12:    for all $(h, s) \in [H] \times \mathcal{S}$ update $\pi_h^{k+1}(\cdot \mid s) \propto \pi_h^k(\cdot \mid s) \cdot \exp(\eta \cdot Q_h^k(s, \cdot))$
13: **end for**
14: **output**: $\pi_{out}$ that is sampled uniformly at random from $\{\pi^k\}_{k \in [K]}$.

---

**Algorithm 2** $\text{OPE}(\pi, \mathcal{D}, r, \mathcal{L}_{\tau^E})$

---

1: Split $\mathcal{D}$ evenly into $H$ disjoint sets $\mathcal{D}_h$ for $h \in [H]$.
2: Set $V_{H+1}(s) \leftarrow 0$ for all $s \in \mathcal{S}$
3: **for** $h = H, \cdots, 1$ **do**
4:    Least-squares regression: $\widehat{f}_h \leftarrow \text{argmin}_{f_h \in \mathcal{F}_h} \sum_{(s_h, a_h, s_{h+1}) \in \mathcal{D}_h} (f_h(s_h, a_h) - V_{h+1}(s_{h+1}))^2$.

5:    **for** $(s, a) \in \mathcal{S} \times \mathcal{A}$ **do**
6:       Compute exploration bonus $b_h(s, a)$ as in (4.4).
7:       $Q_h(s, a) \leftarrow \text{clip}_{[-H, H]}\big(\widehat{f}_h(s, a) + \widetilde{r}_h(s, a) + b_h(s, a)\big)$,
       where $\widetilde{r}_h(s, a) \leftarrow r(s, a) + \lambda \mathcal{L}_{\tau^E, h}(s, a)$.
8:       $V_h(s) \leftarrow \mathbb{E}_{a \sim \pi_h(\cdot | s)}[Q_h(s, a)]$.
9:    **end for**
10: **end for**
11: **output** $\{Q_h\}_{h=1}^H$

---

$\mathcal{S} \times \mathcal{A}$ is upper bounded by $\widetilde{O}(1/\sqrt{N})$, where we omit the dependency of $H$ and generalized Eluder dimension.

After updating the on-policy dataset, in Line 10, the agent can optimize the reward function according to the following loss function:

$$L(\pi, r) = J(\pi^E, r) - J(\pi, r). \tag{4.1}$$

While $L$ is unknown for the learner, in Line 10 of Algorithm 1, we compute an empirical estimate of $L$:

$$\widehat{L}(\pi^k, r) = \frac{1}{n} \cdot \sum_{j=1}^n \sum_{h=1}^H r(s_h^{(j)}, a_h^{(j)}) - \frac{1}{N} \cdot \left[\prod_{h=1}^H \frac{\pi_h^k(a_h \mid s_h)}{\pi_h^{t_k}(a_h \mid s_h)}\right] \cdot \sum_{\tau \in \mathcal{D}^k} \sum_{h=1}^H r(s_h^{(\tau)}, a_h^{(\tau)}), \tag{4.2}$$

where $t_k$ is the index of the last policy which is used to collect fresh data at iteration $k$.

**Assumption 4.1** (Convexity and Lipschitz Continuity of the reward function). We assume that the reward function $r$ is parameterized by a set of parameters $\boldsymbol{\theta} \in \Theta \subset \mathbb{R}^d$ and the estimated loss function $\widehat{L}$ is convex with respect to $\boldsymbol{\theta}$. For any $\boldsymbol{\theta}_1, \boldsymbol{\theta}_2 \in \Theta$, $\|\boldsymbol{\theta}_1 - \boldsymbol{\theta}_2\|_2 = O(1)$, and for any $\boldsymbol{\theta} \in \Theta$, $s, a \in \mathcal{S} \times \mathcal{A}$, $\|\nabla_{\boldsymbol{\theta}} r_{\boldsymbol{\theta}}(s, a)\|_2 = O(1)$.

As a result, in Line 10, we essentially update the reward function $r^k$ by the following projected gradient descent:

$$\boldsymbol{\theta}^{k+1} = \text{proj}_\Theta\big(\boldsymbol{\theta}^k - \eta_{\boldsymbol{\theta}} \cdot \nabla_{\boldsymbol{\theta}}[-\widehat{L}(\pi^k, r^k)]\big). \tag{4.3}$$

To ensure that $\widehat{L}$ can potentially approximate $L$ accurately, we also require the following assumption on the quality of the demonstration trajectories.

**Assumption 4.2** (Quality of the demonstration trajectories). The demonstration data $\tau^E$ satisfies the following property for any reward function $r \in \mathcal{R}$:

$$\left| \frac{1}{n} \sum_{j=1}^{n} \sum_{h=1}^{H} r(s_h^{(j)}, a_h^{(j)}) - J(\pi^E, r) \right| \leq \epsilon_E$$

for some $\epsilon_E > 0$.

Note that the assumption is weak since the error can be bounded by Azuma-Hoeffding inequality.

**Policy optimization module.** The algorithm is a phasic policy optimization algorithm built on Liu et al. (2023b). In Line 12, the policy $\pi_h^{k+1}$ is obtained by taking a mirror descent step from $\pi_h^k$, maximizing the following KL-regularized return:

$$\pi_h^{k+1} = \arg\max_{\pi_h} \eta \langle Q_h^k(s, \cdot), \pi_h(\cdot|s) \rangle - \lambda_{\text{ED}} d_{\text{KL}}(\pi_h | \pi_h^k),$$

where $\lambda_{\text{ED}} \geq 0$ is a regularization parameter, $d_{\text{KL}}$ is the Kullbach-Leibler divergence, and $Q_h^k$ is computed by an optimistic policy evaluation (OPE) subroutine to guide the direction of policy updates.

**Double Exploration in Algorithm 2.** While algorithms widely used in empirical studies, such as PPO, directly use empirical returns to guide the policy updating procedure, we employ Algorithm 2 to estimate the expected return from all state-action pairs.

In OPE (Algorithm 2), we set an exploration bonus according to Lemma C.3,

$$b_h(s, a) = \sqrt{8H^2 \log(H \cdot \mathcal{N}_\mathcal{F}(\epsilon_\mathcal{F})/\delta) + 4\epsilon_\mathcal{F} N + \gamma} \cdot D_{\mathcal{F}_h}((s, a); \mathcal{D}_h), \qquad (4.4)$$

where $D_{\mathcal{F}_h}$ is defined in Definition 4.3 as an exploration bonus to encourage the policy to explore the state-action pairs whose expected return values are hard to predict given the on-policy dataset $\mathcal{D}^k$.

Additionally, we utilize a pretrained uncertainty evaluation function $\mathcal{L}_{\tau^E}$ to explore states beyond the demonstration trajectories. Then, in lines 3-10, we use value iteration to compute the value function considering the sum of the imitation reward $r^k$, the exploration bonus $b_h^k$, and the uncertainty $\mathcal{L}$.

### 4.2 THEORETICAL ANALYSIS

With the ILDE framework introduced in Section 4, we provide a thorough regret analysis for Algorithm 1.

Our analysis addresses a broad category of MDPs, specifically those whose value functions can be effectively approximated and generalized across various state-action pairs. These are referred to as MDPs with bounded generalized Eluder dimension (Agarwal et al., 2023).

**Definition 4.3** (Generalized Eluder dimension, Agarwal et al. 2023). Let $\lambda_{\text{ED}} \geq 0$, and let $\mathbf{Z} = \{z_i\}_{i \in [T]} \in (\mathcal{S} \times \mathcal{A})^{\otimes T}$ be a sequence of state-action pairs . Then the generalized Eluder dimension of a value function class $\mathcal{F} : \mathcal{S} \times \mathcal{A} \to [-H, H]$ with respect to $\lambda_{\text{ED}}$ is defined by $\dim_T(\mathcal{F}) := \sup_{\mathbf{Z}:|\mathbf{Z}|=T} \dim(\mathcal{F}, \mathbf{Z})$,

$$\dim(\mathcal{F}, \mathbf{Z}) := \sum_{i=1}^{T} \min\left(1, D_\mathcal{F}^2(z_i; z_{[i-1]})\right),$$

$$\text{where } D_\mathcal{F}^2(z; z_{[t-1]}) := \sup_{f_1, f_2 \in \mathcal{F}} \frac{(f_1(z) - f_2(z))^2}{\sum_{s \in [t-1]} (f_1(z_s) - f_2(z_s))^2 + \lambda_{\text{ED}}}.$$

We write $\dim_T(\mathcal{F}) := H^{-1} \cdot \sum_{h \in [H]} \dim_T(\mathcal{F}_h)$ for short when $\mathcal{F}$ is a collection of function classes $\mathcal{F} = \{\mathcal{F}_h\}_{h=1}^{H}$ in the context.

**Remark 4.4.** The concept of generalized Eluder dimension was first introduced in Agarwal et al. (2023), where a value-based algorithm was proposed to achieve a nearly optimal regret bound in the online RL setting. Notably, our definition of the function class is somewhat broader, as we apply an unweighted definition of the $D_\mathcal{F}$ distance. Consequently, the definition of generalized Eluder dimension does not require taking the supremum over weights, allowing for a wider class of MDPs.

The analyzed policy optimization method needs a policy evaluation subroutine as introduced in Algorithm 2. To make such a policy evaluation tractable, we make the following realizability assumption.

**Assumption 4.5** (Strong realizability of state-action value function class). For all $h \in H$ and $V_{h+1} : \mathcal{S} \to [-H, H]$, there exists a function $f_h \in \mathcal{F}_h$ such that for all $s \in \mathcal{S}, a \in \mathcal{A}$, we have $f_h(s, a) = \mathbb{P}_h V_{h+1}(s, a)$.

**Definition 4.6** (Covering numbers of function classes). For each $h \in [H]$, there exists an $\epsilon_{\mathcal{F}}$-cover $\mathcal{C}(\mathcal{F}_h, \epsilon_{\mathcal{F}}) \subseteq \mathcal{F}_h$ with size $|\mathcal{C}(\mathcal{F}_h, \epsilon_{\mathcal{F}})| \leq \mathcal{N}(\mathcal{F}_h, \epsilon_{\mathcal{F}})$, such that for any $f \in \mathcal{F}$, there exists $f' \in \mathcal{C}(\mathcal{F}_h, \epsilon_{\mathcal{F}})$, such that $\|f - f'\|_\infty \leq \epsilon_{\mathcal{F}}$. For any $\epsilon_{\mathcal{F}} > 0$, we define the uniform covering number of $\mathcal{F}$ with respect to $\epsilon_{\mathcal{F}}$ as $\mathcal{N}_{\mathcal{F}}(\epsilon_{\mathcal{F}}) := \max_{h \in [H]} \mathcal{N}(\mathcal{F}_h, \epsilon_{\mathcal{F}})$.

**Theorem 4.7** (Regret bound for Algorithm 1). Suppose Assumptions 4.5, 4.1 and 4.2 hold. if we set $\gamma = H^2$, $\epsilon_{\mathcal{F}} = 1/N$, $\eta = \sqrt{\log |\mathcal{A}|}/H\sqrt{K}$, $m = \sqrt{K}/H\sqrt{\log |\mathcal{A}|}$, $\eta_{\boldsymbol{\theta}} = O(1/\sqrt{H^2 K})$, and $N = KH \dim_T(\mathcal{F}) \log \mathcal{N}_{\epsilon_{\mathcal{F}}}(\mathcal{F})/\sqrt{\log |\mathcal{A}|}$, where $K = \left( \frac{T}{H^2 \dim_T(\mathcal{F}) \log \mathcal{N}_{\epsilon_{\mathcal{F}}}(\mathcal{F})} \right)^{2/3}$, then with probability at least $1 - \delta$, Algorithm 1 yields a regret of

$$\widetilde{O} \left( H^{8/3} \big( \dim_T(\mathcal{F}) \log \mathcal{N}_{\epsilon_{\mathcal{F}}}(\mathcal{F}) \big)^{1/3} T^{2/3} + \epsilon_E T \right). \tag{4.5}$$

**Remark 4.8.** This is the first theoretical guarantee for imitation learning in MDPs with nonlinear function approximation. As shown in Zhao et al. (2023), the considered setting captures MDPs with bounded Eluder dimension (Russo & Van Roy, 2013) and thus also captures linear MDPs (Jin et al., 2020) as special cases. If we select a policy from $\{\pi^k\}_{k=1}^K$ uniformly at random, the expected loss for that policy is $O(\epsilon + \epsilon_E)$ when $T = \widetilde{\Theta}(H^8 \dim_T(\mathcal{F}) \log \mathcal{N}_{\epsilon_{\mathcal{F}}}(\mathcal{F})/\epsilon^3)$, which provides a sample complexity bound for Algorithm 1.

## 5 PRACTICAL IMPLEMENTATION OF ILDE

---
**Algorithm 3** Imitation Learning with Double Exploration (ILDE)

---
1: **input**: number of episodes $T$, period of collecting fresh data $m$, batch size $N$, learning rate $\eta$, demonstration trajectories $\tau^E$, reward function $r^1$, pretrained loss function $\mathcal{L}_{\tau_E}$.
2: **for** $t = 1, \ldots, T$ **do**
3:     Sample trajectories $\mathcal{D}^t$ from $\pi^t$.
4:     Update the imitation reward $r^t$ by minimizing the loss of the discriminator trained to distinguish between the demonstration trajectories and $\mathcal{D}^t$ (e.g. (5.1)).
5:     Compute the aggregated reward $\widetilde{r}^t(s, a) = r^t(s, a) + \lambda \mathcal{L}_{\tau_E}(s, a)$ for all state-action pairs $(s, a)$ in $\mathcal{D}^t$.
6:     Update the current policy $\pi^t$ via policy optimization method (e.g., PPO) with reward $\widetilde{r}^t(s, a) + b(s, a)$ where $b(s, a)$ is the exploration bonus and obtain $\pi^{t+1}$.
7: **end for**

---

While Algorithm 1 provides a solid sample-complexity guarantee, it lacks computational efficiency and is challenging to implement at scale. In this section, we focus on the crucial elements of Algorithm 1 and introduce the proposed Imitation Learning with Double Exploration (ILDE) framework. The algorithm, summarized in pseudocode in Algorithm 3, includes the following key modules:

- Imitation reward module: This module minimizes the loss of a discriminator which distinguishes between demonstration data $\tau^E$ and data obtained during policy optimization $\mathcal{D}^t$. Theoretically, the imitation reward function is progressively optimized according to the gap between the average return of demonstration trajectories and that of current trajectories. In practice, to implement the module, we maintain a discriminator $D_\theta$ which continue minimizing the following loss:

$$\mathbb{E}_{(s,a) \sim \tau^E} \big[ \log D_\theta(s, a) \big] + \mathbb{E}_{(s,a) \sim \pi^t} \big[ \log(1 - D_\theta(s, a)) \big] + \beta \mathbb{E}_{s \sim \pi^t} \big[ d_{\mathrm{KL}}(E'(z|s) | r^t) - I_c \big], \quad (5.1)$$

  where encoder $E'$ is introduced to incorporate a variational discriminator bottleneck (Peng et al., 2019) to improve GAIL, $\beta$ is the scaling weight, and $I_c$ is the information constraint. Further details can be found in Appendix B.2.1.

- Pretrained uncertainty evaluation module: To implement this module, we make use of GIRIL. That is, we pretrain a VAE model to estimate the transition kernel $\widehat{\mathbb{P}}$ over the demonstration trajectories

$\tau^E$ and compute the intrinsic reward $\mathcal{L}_{\tau^E, h}(s, a)$ (3.4) for all $(s, a)$ in $\mathcal{D}^t$. Further details can be found in Appendix B.2.2.

- Exploration bonus $b$: To analyze the ILDE framework theoretically in Section 4, we choose the $D_\mathcal{F}$-uncertainty of state-action pairs as the exploration bonus, which reflects the uncertainty of a state-action pair after collecting enough data using the current policy. To efficiently implement the exploration bonus in practice, we utilize state entropy (Seo et al., 2021) in a representation space of a feature extractor $f$. We utilize a $k$-nearest neighbor entropy estimator to calculate the exploration bonus as $b(s, a) = \log(||y - y^{k-\mathrm{NN}}||_2 + 1)$, where $y = f(s)$ is a state representation from $f$ and $y^{k-\mathrm{NN}}$ is the $k$-nearest neighbor of $y$ within a set of $N$ representations $\{y_1, y_2, \cdots, y_N\}$, where $N$ is the batch size. The state-dependency rather than state-action dependency of the bonus is motivated by the technique of entropy regularisation in the policy optimization already encouraging spread over the action space. Further details can be found in Appendix B.2.3.

- Policy optimization module: While the above modules provide the design of the reward, the policy optimization module optimises the policy for the objective. For policy optimization, we use an actor-critic variant of proximal policy optimization (PPO) (Schulman et al., 2017) with generalized advantage updating.

## 6 EXPERIMENTS

We now conduct experiments based on four key hypotheses. First, we hypothesize that pure imitation learning methods such as VAIL will fail with limited demonstrations. Second, we hypothesize that a pre-trained model-based exploration reward such as GIRIL alone provides a stable improvement but may not reach a wide set of states due to the curiosity bias preferring certain states. Third, the state entropy reward bonus provides an exploration reward which provides a high reachability across a wide variety of states, thereby improving sample efficiency and overall performance. Fourth, VAIL's imitation reward can provide additional performance gains compared to pure exploration (i.e. exploration bonus and/or curiosity reward) when the imitation reward is reliable while there is no effect when the imitation reward is unreliable.

### 6.1 ATARI GAMES

To test these hypotheses, we compare our proposed ILDE to VAIL, GIRIL, and ablations on six Atari games within OpenAI Gym (Brockman et al., 2016) in a challenging setting where the agent receives even less demonstration data than in so-called one-life demonstration data (Yu et al., 2020). The imitation learning agents are trained according to the design of the imitation reward and/or any potential bonuses but are evaluated based on their actual scores on the Atari games. We summarize more experimental details in the Appendix B.

**Demonstrations.** Our setting is very challenging compared to existing work because the demonstration data are very sparse, since they are based on partial one-life demonstrations. A one-life demonstration only contains the states and actions performed by an expert player until they die for the first time in a game. We generate the one-life demonstrations using a PPO agent trained with ground-truth rewards for 10 million simulation steps. Table 4 in the Appendix B.1 shows that a one-life demonstration contains much less data than a full-episode demonstration. To make the problem even more challenging, the imitation learning agents receive demonstrations which comprise only 10% of a full one-life demonstration.

**Baselines.** We compare our ILDE with the following baselines: (1) VAIL (Peng et al., 2019), an improved GAIL variant using variation discriminator bottleneck, which updates the policy based on $r^k$ only; (2) GIRIL (Yu et al., 2020), the generative intrinsic reward-driven imitation learning method; (3) ILDE w/o $b$, an ablation of ILDE without the exploration bonus $b$; and (4) ILDE w/o $r^k$, an ablation of IDLE without the imitation reward $r^k$.

**Results.** Table 1 compares the performance of our ILDE and other baselines. ILDE outperforms the other baselines in terms of achieving higher average returns and outperforming experts in more games. Notably, ILDE achieves an average performance that is 5.33 times that of the expert demonstrator across the 6 Atari games. More specifically, ILDE outperforms the expert in all of the 6 Atari games and outperforms GIRIL in 4 games. The ablation study in Table 1 indicates the significance of each reward component. ILDE achieves much higher sample efficiency improvements than GIRIL, as shown in Table 10 and 11 of Appendix B.2.5. Further, we also show in Appendix B.2.7 that ILDE

Table 1: Average return of Expert demonstrator, VAIL, GIRIL, ILDE w/o $\lambda\mathcal{L}_{\tau_E}$, ILDE w/o $b$, ILDE w/o $r^k$ and ILDE with one-life demonstration data on six Atari games. The results are reported in the format of $\mathrm{mean} \pm \mathrm{std}$ over 10 random seeds with better-than-GIRIL performance in bold. "#Games>Expert" denotes the number and ratio of games that an imitation learning method outperforms the expert. "Improve vs Expert" denotes the average performance improvements of IL methods versus the expert across 6 Atari games.

| Atari Games | Expert | VAIL ($r^k$) | GIRIL | ILDE w/o $\lambda\mathcal{L}_{\tau_E}$ | ILDE w/o $b$ | ILDE w/o $r^k$ | ILDE |
|---|---|---|---|---|---|---|---|
| BeamRider | 1,918±645 | 91±116 | 8,524±1,085 | 3,233±3,340 | 8,521±1,234 | **11,453±2,416** | **11,453±2,319** |
| DemonAttack | 8,820±2,893 | 979±863 | 72,381±19,851 | 143±120 | **78,513±11,553** | 60,931±10,376 | 64,498±13,294 |
| BattleZone | 22,410±5,839 | 5,081±4,971 | 25,203±22,868 | 5,412±1,816 | 18,179±21,181 | **43,708±22,733** | **59,109±25,575** |
| Qbert | 6,246±2,964 | 3,754±4,460 | 76,491±81,265 | 10,849±5,018 | 26,900±52,146 | 54,169±68,305 | 70,566±75,310 |
| Krull | 6,520±941 | 7,937±8,183 | 20,935±23,796 | 857±916 | 16,715±23,153 | **27,495±28,773** | **23,716±16,790** |
| StarGunner | 22,495±11,516 | 219±182 | 542±188 | 764±455 | 478±239 | **16,526±32,946** | **25,361±38,557** |
| #Games>Expert | 0 / 0% | 1 / 16.7% | 5 / 83.3% | 2 / 33.3% | 4 / 66.7% | 5 / 83.3% | **6 / 100%** |
| Improve vs Expert | 1.0 | 0.37 | 4.87 | 0.64 | 3.50 | 4.71 | **5.33** |

is robust to noise in the demonstrations. Last, we note that the limited demonstration setting is challenging for IQ-Learn (see Table 12 in Appendix B.2.6).

## 6.2 CONTINUOUS CONTROL TASKS

Additionally, we also conduct experiments on the MuJoCo environments. Table 2 compares the performance of ILDE and other baselines. ILDE consistently outperforms other baselines in the four continuous control tasks by achieving higher average returns over 10 random seeds. We summarized more experimental details for MuJoCo experiments in Appendix B.2.8. Based on our current experiments, we observed that ILDE performs exceptionally well in games, while its improvement on MuJoCo is less significant. This may suggest that the current design of the intrinsic reward is better suited for taskswhere the extrinsic reward is more aligned with the objective of "seeking novelty".This may also indicate that our algorithm does not (only) extrapolate the intention in the partial demonstration. Additional experiments (see Table 13 in Appendix B.2.6) show the potential of ILDE to generalize to other adversarial IL methods.

Table 2: Average return of VAIL, GIRIL, and ILDE on the MuJoCo tasks. The average returns are calculated on the last 4 evaluation points. The results are reported in the format of $\mathrm{mean} \pm \mathrm{std}$ over 10 random seeds with the best performance in bold.

| Tasks | VAIL ($r^k$) | GIRIL | ILDE |
|---|---|---|---|
| Reacher | -499±183 | -60±24 | **-51±15** |
| Hopper | 1,264±783 | 1,447±273 | **1,878±690** |
| Walker2d | 879±825 | 997±575 | **998±694** |
| HumanoidStandup | 52,635±6,370 | 76,134±10,826 | **77,887±6,123** |

## 7 CONCLUSION

Achieving beyond-expert performance and improving sample complexity in complicated tasks are two crucial challenges in imitation learning. To address these difficulties, we propose Imitation Learning with Double Exploration (ILDE), which optimizes an uncertainty-regularized discrepancy that combines the distance to the expert policy with cumulative uncertainty during exploration. Theoretically, we propose ILDE-NPG with a theoretical regret guarantee, a first for imitation learning in MDPs with nonlinear function approximation. Empirically, we introduce a flexible ILDE framework with efficient exploration strategies and demonstrate ILDE's outstanding performance over existing baselines in average return and sample complexity. It is also worth noting that our analysis cannot be directly applied to the practical version implemented in experiments.

**Limitation of our work.** Our method assumes that the target task to be imitated is aligned with the concept of seeking novelty or curiosity. Its ability to achieve performance beyond the demonstration is not only due to the algorithm's capacity to understand and extrapolate the intention behind the partial demonstration but also due to an alignment between the extrinsic reward and seeking novelty.

ACKNOWLEDGMENTS

We thank the anonymous reviewers and area chair for their helpful comments. HZ and QG are supported in part by the NSF grants CPS-2312094, IIS-2403400 and Sloan Research Fellowship. HZ is also supported by UCLA-Amazon Science Hub Fellowship. The views and conclusions contained in this paper are those of the authors and should not be interpreted as representing any funding agencies. XY, DB, and IWT are supported by CFAR, Agency for Science, Technology and Research, Singapore. This project is supported by the National Research Foundation, Singapore and Infocomm Media Development Authority under its Trust Tech Funding Initiative. Any opinions, findings and conclusions or recommendations expressed in this material are those of the authors and do not reflect the views of National Research Foundation, Singapore and Infocomm Media Development Authority.

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

## A    MOTIVATION TABLE

Table 3 illustrates the strengths of each component in ILDE, which helps to explain the motivation of ILDE.

Table 3: Comparison of three reward components with three attributes.

|  | Stability | Exploration | Imitation |
|---|---|---|---|
| GIRIL reward | ++ | + | + |
| Bonus $b$ | + | ++ | - |
| VAIL $r^k$ | - | - | ++ |

## B    AUXILIARY EXPERIMENTAL DETAILS

### B.1    DEMONSTRATION LENGTHS IN THE ATARI ENVIRONMENT

Table 4 compares the lengths of one-life demonstrations and full-episode demonstrations in Atari games. In the experiments, we only use 10% of the one-life demonstrations for imitation learning, which is more challenging than previous work in learning with limited demonstrations (Yu et al., 2020).

### B.2    EXPERIMENTAL SETUP AND ADDITIONAL RESULTS

This subsection presents details of VAIL, GIRIL, state entropy bonus, experimental setup, and additional results.

#### B.2.1    DETAILS OF THE VARIATIONAL ADVERSARIAL IMITATION LEARNING (VAIL)

Generative adversarial imitation learning (GAIL) (Ho & Ermon, 2016) regards imitation learning as a distribution matching problem, and updates policy using adversarial learning (Goodfellow et al., 2020).

Table 4: Demonstration lengths in the Atari environment.

| Atari Games | One-life Demo. | | Full-episode Demo. | |
|---|---|---|---|---|
| | Length | # Life | Length | # Life |
| Beam Rider | 702 | 1 | 1,412 | 3 |
| Demon Attack | 2,004 | 1 | 11,335 | 4 |
| Battle Zone | 260 | 1 | 1,738 | 5 |
| Q*bert | 787 | 1 | 1,881 | 4 |
| Krull | 662 | 1 | 1,105 | 3 |
| Star Gunner | 118 | 1 | 2,117 | 5 |

Previous work has demonstrated that GAIL does not work well in high-dimensional environments such as Atari games (Brown et al., 2019). Variational adversarial imitation learning (VAIL) (Peng et al., 2019) improves GAIL by compressing the information via a variational information bottleneck (VDB). VDB constrains the information flow in the discriminator using an information bottleneck. By enforcing a constraint on the mutual information between the observations and the discriminator's internal representation, VAIL significantly outperforms GAIL by optimizing the following objective:

$$\min_{D_\theta, E'} \max_{\beta \geq 0} \mathbb{E}_{(s,a)\sim\tau^E}\Big[\mathbb{E}_{z\sim E'(z|s)}\big[\log(-D_\theta(z))\big]\Big] + \mathbb{E}_{(s,a)\sim\pi^k}\Big[\mathbb{E}_{z\sim E'(z|s)}\big[-\log(1-D_\theta(z))\big]\Big]$$
$$+ \beta\mathbb{E}_{s\sim\widetilde{\pi}}\big[d_{\text{KL}}(E'(z|s)|r^k) - I_c\big],$$
(B.1)

where $\widetilde{\pi} = \frac{1}{2}\pi^E + \frac{1}{2}\pi^k$ represents a mixture of the expert policy and the agent's policy, $E'$ is the encoder for VDB, $\beta$ is the scaling weight, and $I_c$ is the information constraint. The reward for $\pi$ is then specified by the discriminator $r_t = -\log\big(1 - D_\theta(\boldsymbol{\mu}_{E'}(s))\big)$.

### B.2.2 DETAILS OF THE GENERATIVE INTRINSIC REWARD DRIVEN IMITATION LEARNING (GIRIL)

Previous inverse reinforcement learning (IRL) methods usually fail to achieve expert-level performance when learning with limited demonstrations in high-dimensional environments. To address this challenge, Yu et al. (2020) proposed generative intrinsic reward-driven imitation learning (GIRIL) to empower the agent with the demonstrator's intrinsic intention and better exploration ability. This was achieved by training a novel reward model to generate intrinsic reward signals via a generative model. Specifically, GIRIL leverages a conditional VAE (Sohn et al., 2015) to combine a backward action encoding model and a forward dynamics model into a single generative model. The module is composed of several neural networks, including recognition network $q_\phi(z|s_t, s_{t+1})$, a generative network $p_\theta(s_{t+1}|z, s_t)$, and prior network $p_\theta(z|s_t)$. GIRIL refers to the recognition network (i.e. the probabilistic *encoder*) as a backward action encoding model, and the generative network (i.e. the probabilistic *decoder*) as a forward dynamics model. Maximizing the following objective to optimize the module:

$$J(p_\theta, q_\phi) = \mathbb{E}_{q_\phi(z|s_t, s_{t+1})}[\log p_\theta(s_{t+1}|z, s_t)] - d_{\text{KL}}(q_\phi(z|s_t, s_{t+1})\|p_\theta(z|s_t))$$
$$- \alpha d_{\text{KL}}(q_\phi(\widehat{a}_t|s_t, s_{t+1})|\pi_E(a_t|s_t))$$
(B.2)

where $z$ is the latent variable, $\pi_E(a_t|s_t)$ is the expert policy distribution, $\widehat{a}_t = \text{Softmax}(z)$ is the transformed latent variable, $\alpha$ is a positive scaling weight. The reward model will be pre-trained on the demonstration data and used for inferring intrinsic rewards for the policy data. The intrinsic reward is calculated as the reconstruction error between $\widehat{s}_{t+1}$ and $s_{t+1}$:

$$r_t = \|\widehat{s}_{t+1} - s_{t+1}\|_2^2$$
(B.3)

where $\|\cdot\|_2$ denotes the L2 norm, and $\widehat{s}_{t+1} = \text{decoder}(a_t, s_t)$.

### B.2.3 DETAILS OF STATE ENTROPY BONUS

State entropy maximization has been demonstrated to be a simple and compute-efficient method for exploration (Seo et al., 2021). The key idea for this method to work in a high-dimensional environment is to utilize a $k$-nearest neighbor state entropy estimator in the state representation space.

$k$-**nearest neighbor entropy estimator.** Let $X$ be a random variable with a probability density function $p$ whose support is a set $\mathcal{X} \subset \mathbb{R}^d$. Then its differential entropy is given by

$$\mathcal{H}(X) = - \int_{\mathcal{X}} p(x) \log p(x) dx.$$

Since estimating $p$ is not tractable for high-dimensional data, a particle-based $k$-nearest neighbors ($k$-NN) entropy estimator (Singh et al., 2003) can be used:

$$\mathcal{H}(X) \approx \frac{1}{N} \sum_{i=1}^{N} \log \frac{N \cdot \|x_i - x_i^{k-\text{NN}}\|_2^d \cdot \widehat{\pi}^{\frac{d}{2}}}{k \cdot \Gamma(\frac{d}{2} + 1)} + C_k$$

$$\propto \frac{1}{N} \sum_{i=1}^{N} \log \|x_i - x_i^{k-\text{NN}}\|_2, \tag{B.4}$$

where $x_i^{k-\text{NN}}$ is the $k$-nearest neighbor of $x_i$ within a set of $N$ representations $\{x_1, x_2, \cdots, x_N\}$, $\Gamma$ is the gamma function, $\widehat{\pi}$ refers to an estimate of the number $\pi$ (as opposed to the policy), and $\log(k - \psi)$ where $\psi$ is the digamma function.

**State entropy estimate as bonus.** Following Seo et al. (2021), we define the bonus to be proportional to the state entropy estimate in (B.4),

$$b(s_i) := \log(\|y_i - y_i^{k-\text{NN}}\|_2 + 1), \tag{B.5}$$

where $y_i = f(s_i)$ is a fixed representation from a state feature extractor $f$ and $y_i^{k-\text{NN}}$ is the $k$-nearest neighbor of $y_i$ within a set of $N$ representations $\{y_1, y_2, \cdots, y_N\}$. The use of a fixed representation space produces a more stable intrinsic reward since the distance between a given pair of states does not change during training (Seo et al., 2021). In our implementation, we use the identity function as the feature extractor $f$.

### B.2.4 EXPERIMENTAL SETUP

For the GIRIL and ILDEs, our first step was to train a reward model for each game using the 10% of a one-life demonstration. The training was conducted with the Adam optimizer at a learning rate of 3e-4 and a mini-batch size of 32 for 1,000 epochs. In each training epoch, we sampled a mini-batch of data every four states. To evaluate the quality of our learned reward, we used the trained reward learning module to produce intrinsic rewards for policy data and trained a policy to maximize the inferred intrinsic rewards via PPO. For VAIL, we trained the discriminator using the Adam optimizer with a learning rate of 3e-4. The discriminator was updated at every policy step. We trained the PPO policy for all imitation learning methods for 50 million steps. We compare ILDE with baselines in the measurement of average return and sample efficiency. The average return was measured using the true rewards in the environment. We measured the sample efficiency based on the step after which an imitation learning method can continuously outperform the expert until the training ends. All the experiments are executed in an NVIDIA A100 GPU with 40 GB memory.

**Network architectures.** We implement our ILDE and all baselines in the feature space of a state feature extractor $f$ with 3 convolutional layers. The dimension of the state feature is 5184. In the state feature space, we use 3-layer MLPs to implement the discriminator for VAIL and the encoder and decoder for GIRIL. For a fair comparison, we used an identical policy network for all methods. We used the actor-critic approach for training PPO policy for all imitation learning methods. Table 5 shows the architecture details of the state feature extractor and the actor-critic network. Table 6 shows the MLP architectures, i.e., GIRIL's *encoder* and *decoder* and VAIL's discriminator.

**Hyperparameter settings.** We implement all imitation learning methods and experiments based on the GitHub repository by Kostrikov (2018). Additional parameter settings are explained in Table 7. Note that $\lambda$, which controls the curiosity bonus was set without tuning to $\lambda = 10$ in our experiments. We use the same hyperparameter setting for the different experiments within a domain (Atari vs MuJoCo), apart from a multiplicative constant based on the range of the curiosity. However, to further explore its effect, we perform a parametric study with $\lambda$ ranging in $\{20, 10, 5, 2\}$. The results in Table 8 show that ILDE's performance slightly increases with decreasing $\lambda$.

Table 5: Architectures of state feature extractor and actor-critic network for Atari games.

| State feature extractor | Actor-Critic network | |
|---|---|---|
| $4 \times 84 \times 84$ States | $4 \times 84 \times 84$ States | |
| $3 \times 3$ conv, 32 LeakyReLU | $8 \times 8$ conv, 32, stride 4, ReLU | |
| $3 \times 3$ conv, 32 LeakyReLU | $4 \times 4$ conv, 64, stride 2, ReLU | |
| $3 \times 3$ conv, 64 LeakyReLU | $3 \times 3$ conv, 32, stride 1, ReLU | |
| flatten $64 \times 9 \times 9 \rightarrow 5184$ | dense $32 \times 7 \times 7 \rightarrow 512$ Categorical Distribution | dense $32 \times 7 \times 7 \rightarrow 1$ |
| State features | Actions | Values |

Table 6: Architectures of MLP-based GIRIL's encoder and decoder, and VAIL's discriminator.

| GIRIL | | VAIL |
|---|---|---|
| encoder | decoder | discriminator |
| 5184 State features and next-state features | Actions and 5184 State features | Actions and 5184 State features |
| dense $5184 \times 2 \rightarrow 1024$, LeakyReLU dense $1024 \rightarrow 1024$, LeakyReLU dense $1024 \rightarrow \mu$, dense $1024 \rightarrow \sigma$ reparameterization $\rightarrow$ #Actions | dense $5182 +$#Actions $\rightarrow 1024$, LeakyReLU dense $1024 \rightarrow 1024$, LeakyReLU dense $1024 \rightarrow 5184$ | dense $5182 \rightarrow 1024$, LeakyReLU dense $1024 \rightarrow 1024$, LeakyReLU split 1024 into 512, 512 dense $512 \rightarrow \mu_{E'}$, dense $512 \rightarrow \sigma_{E'}$ reparameterization $\rightarrow 1$ |
| Latent variables | 5184 Predicted next state features | $\mu_{E'}, \sigma_{E'}$, discrimination |

Table 7: Hyperparameter settings for Atari games.

| Parameter | Setting |
|---|---|
| Actor-critic learning rate | 2.5e-4, linear decay |
| PPO clipping | 0.1 |
| Reward model learning rate | 3e-4 |
| Discount factor | 0.99 |
| GAE Lambda | 0.95 |
| Critic loss coefficient | 0.5 |
| Entropy regularisation | 0.01 |
| Epochs per batch | 4 |
| Batch size | $32 \times 128$ (parallel workers $\times$ time steps) |
| Mini batch size | 32 |
| Evaluation frequency | every 200 policy updates |
| GIRIL objective $\alpha$ | 100.0 |
| Mini batch size for training the reward model | 32 |
| Total training epoch of reward model | $1,000$ |
| VAIL's bottleneck loss weight $\beta$ | 1.0 |
| Information constraint $I_c$ | 0.2 |
| $\lambda$ in Algorithm 3 | 10.0 |

Table 8: The effect of $\lambda$ on ILDE performance in the BeamRider task.

| 20 | 10 | 5 | 2 |
|---|---|---|---|
| $8,973 \pm 1,315$ | $11,453 \pm 2,319$ | $11,947 \pm 1,228$ | $12,600 \pm 1,351$ |

### B.2.5 ADDITIONAL RESULTS

To better illustrate the significance of the improvement of ILDE against the expert demonstrator, we normalize the results by stating that the expert was 1.0. The normalized results are summarized in Table 9. On average, VAIL is worse than the expert, only outperforming the expert in Krull. GIRIL

archives a performance that is 4.87 times the expert, performing the best in Q*bert. ILDE is the best, achieving a performance that is 5.33 times the expert. Figure 1 illustrates the learning curves of imitation learning methods in the 6 Atari games over 10 random seeds.

Table 9: Average performance improvements of imitation learning methods versus the expert demonstrator. The expert performance is regarded as 1.0. "Improve vs Expert" denotes the average improvements of IL methods versus the expert across 6 Atari games.

| Atari Games | Expert | VAIL ($r^k$) | GIRIL | ILDE w/o $b$ | ILDE w/o $r^k$ | ILDE |
|---|---|---|---|---|---|---|
| BeamRider | 1.0 | 0.05 | 4.44 | 4.44 | 5.97 | 5.97 |
| DemonAttack | 1.0 | 0.11 | 8.21 | 8.90 | 6.91 | 7.31 |
| BattleZone | 1.0 | 0.23 | 1.12 | 0.81 | 1.95 | 2.64 |
| Q*bert | 1.0 | 0.60 | 12.25 | 4.31 | 8.67 | 11.30 |
| Krull | 1.0 | 1.22 | 3.21 | 2.56 | 4.22 | 3.64 |
| StarGunner | 1.0 | 0.01 | 0.02 | 0.02 | 0.73 | 1.13 |
| Improve vs Expert | 1.0 | 0.37 | 4.87 | 3.50 | 4.71 | **5.33** |

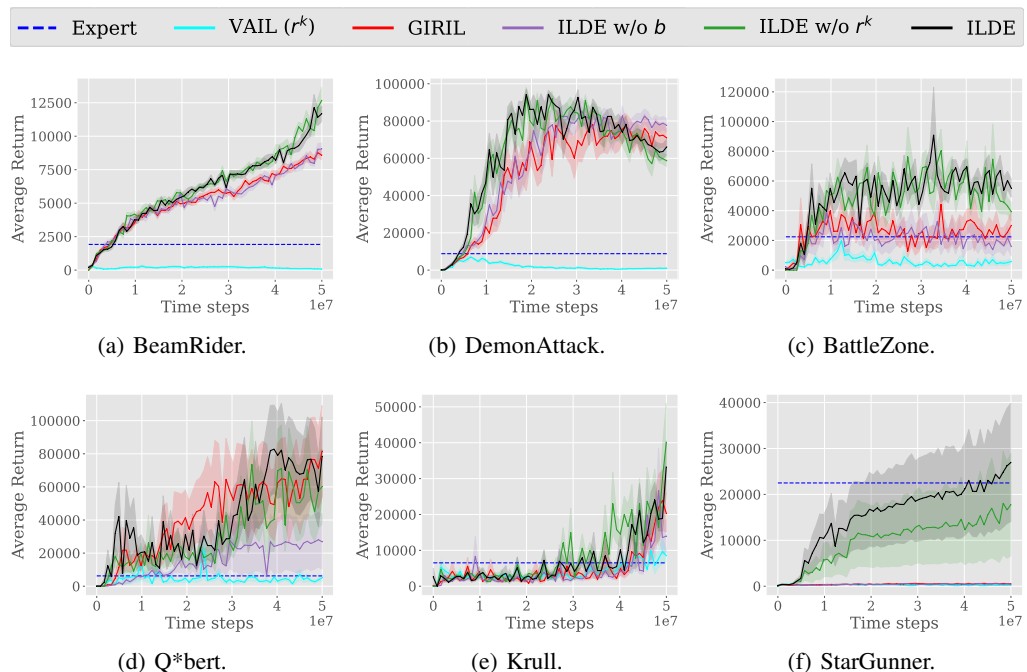

Figure 1: Average return vs. number of simulation steps on Atari games. The solid lines show the mean performance over 10 random seeds. The shaded area represents the standard deviation from the mean. The blue dotted line denotes the average return of the expert demonstrator. The area above the blue dotted line means performance beyond the expert.

Additionally, we summarize the sample efficiency improvements of VAIL and ILDE methods versus GIRIL in Table 11. The sample efficiency of VAIL in outperforming experts is much worse than GIRIL and ILDE variants. ILDE shows impressive sample efficiency improvements across several games, and especially so in BattleZone, where the improvement over GIRIL is 90.06%.

### B.2.6 EXPERIMENTS WITH IQ-LEARN AND HYPE

As additional baseline comparisons, we include IQ-Learn (Garg et al., 2021) and HyPE (Ren et al., 2024).

First, we compare IQ-Learn on our Atari task setting (with 10% of a one-life demonstration). The results in Table 12 show that IQ-Learn does not perform well on Atari tasks, likely because the

Table 10: The sample efficiency of imitation learning methods. The metric for sample efficiency is based on the ratio $t/T$, where $t$ is the first time step such that the agent achieves at least expert performance continuously within the interval $[t, T]$. In Atari, $T$ is 50 million time steps.

| Atari Games | VAIL ($r^k$) | GIRIL | ILDE w/o $b$ | ILDE w/o $r^k$ | ILDE |
|---|---|---|---|---|---|
| BeamRider | - | 9.84% | **8.20%** | 9.84% | 11.48% |
| DemonAttack | - | 13.12% | **9.84%** | **11.48%** | **8.20%** |
| BattleZone | - | 98.31% | - | **11.48%** | **9.84%** |
| Qbert | - | 9.84% | 21.31% | **6.56%** | **8.20%** |
| Krull | 96.67% | 85.20% | 91.76% | **72.10%** | **83.57%** |
| StarGunner | - | - | - | - | **93.40%** |

Table 11: Sample efficiency improvements of VAIL and ILDE methods versus the GIRIL, defined as $\frac{t_G - t_X}{t_G}$, where $t_G$ is the time for GIRIL to reach expert-level and $t_X$ is the time for the method of comparison to reach expert-level (continuously until time $T$). "Improve vs GIRIL" denotes the average improvements of IL methods versus the GIRIL across 6 Atari games. To make the scores feasible, we limit the maximum of $t_X$ to $T$.

| Atari Games | VAIL ($r^k$) | GIRIL | ILDE w/o $b$ | ILDE w/o $r^k$ | ILDE |
|---|---|---|---|---|---|
| BeamRider | -916.26% | 0% | 16.67% | 0% | -16.67% |
| DemonAttack | -662.21% | 0% | 25% | 12.5% | 37.5% |
| BattleZone | -1.71% | 0% | -1.72% | 88.32% | 90.06% |
| Q*bert | -916.26% | 0% | -116.57% | 33.33% | 16.67% |
| Krull | -13.46% | 0% | -7.69% | 15.37% | 1.91% |
| StarGunner | 0% | 0% | 0% | 0% | 6.6% |
| Average | -418.32% | 0% | -14.05% | 24.92% | 22.68% |

demonstration data are extremely limited. We do not include HyPE in the Atari tasks because the HyPE implementation does not support Atari environments.

Table 12: Performance of IQ-Learn on Atari benchmarks.

| Atari Game | IQ-Learn performance |
|---|---|
| BeamRider | $0.0 \pm 0.0$ |
| DemonAttack | $88.5 \pm 92$ |
| BattleZone | $3,800 \pm 870$ |
| Qbert | $225 \pm 125$ |
| Krull | $4,500 \pm 2,030$ |
| StarGunner | $200 \pm 100$ |

Second, we compare HyPE and HyPE-ILDE, a variant of our framework implemented with HyPE, on two MuJoCo tasks based on only a single demonstration (which is significantly more challenging than the 64 demonstrations as in Ren et al. (2024)). The results in Table 13 show that HyPE-ILDE clearly outperforms HyPE.

Table 13: Comparison of HyPE and HyPE-ILDE on MuJoCo tasks.

| MuJoCo Tasks | HyPE | HyPE-ILDE |
|---|---|---|
| Ant-v3 | $785.8 \pm 528.6$ | $\mathbf{813.8 \pm 833.0}$ |
| Hopper-v3 | $2,995.0 \pm 745.4$ | $\mathbf{3,372.6 \pm 359.4}$ |

### B.2.7 ILDE FOR NOISY DEMONSTRATIONS

To test the robustness of ILDE, we generate noisy demonstrations by injecting noise into the environment in the following manner: with probability $p_{\text{tremble}}$, a random action is executed instead of the one chosen by the policy (Ren et al., 2024). Due to the limited time and computational resources, we test the robustness of ILDE for noisy demonstrations on BeamRider without tuning any hyperparameters. The results in Table 14 show that ILDE consistently outperforms the expert (1,918,645) and GIRIL (8,5241,085) on the noisy demonstrations with noise rate of $p_{\text{tremble}} \in \{0.01, 0.05, 0.1, 0.3, 0.5\}$, providing insights into ILDE's robustness in real-world settings.

Table 14: The effect of noise rate ($p_{\text{tremble}}$) on ILDE performance for noisy demonstrations on the BeamRider task.

| 0.0 | 0.01 | 0.05 | 0.1 | 0.3 | 0.5 |
|-----|------|------|-----|-----|-----|
| $11,453 \pm 2,319$ | $12,353 \pm 2,753$ | $8,893 \pm 951$ | $9,935 \pm 1,952$ | $9,728 \pm 1,847$ | $10,735 \pm 1,198$ |

### B.2.8 CONTINUOUS CONTROL TASKS

Except for the above evaluation on the Atari games with high-dimensional state space and discrete action space, we also evaluated our method on continuous control tasks where the state space is low-dimensional and the action space is continuous. The continuous control tasks were from MuJoCo (Todorov et al., 2012).

**Demonstrations.** We use the demonstrations from the open-source repository "pytorch-a2c-ppo-acktr-gail" (Kostrikov, 2018). We compare ILDE with VAIL and GIRIL on the continuous control tasks (i.e., Reacher, Hopper, Walker2d, and HumanoidStandup) with 4 trajectories in the demonstration for each task.

**Experimental setups.** Similar to the setups in Atari games, we need to pretrain a reward model for GIRIL and ILDE using the demonstrations. The training is conducted with the Adam optimizer at a learning rate of 3e-4 and a mini-batch size of 32 for 10,000 epochs. In each training epoch, we sample a mini-batch of data every 20 states. To evaluate the quality of our learned reward, we use the trained reward learning module to produce intrinsic rewards for policy data and trained a policy to maximize the inferred intrinsic rewards via PPO. For VAIL, we train the discriminator using the Adam optimizer with a learning rate of 3e-4. The discriminator is updated at every policy step. We train the PPO policy for all imitation learning methods for 10 million steps. We compare ILDE with baselines in the measurement of average return. The average return is measured using the true rewards in the environment. All the experiments are executed in an NVIDIA A100 GPU with 40 GB memory.

**Network architectures.** We use 3-layer MLPs to implement the discriminator for VAI, and the encoder and the decoder for GIRIL and ILDE. For a fair comparison, we used an identical policy network for all methods. We used the actor-critic approach for training PPO policy for all the imitation learning methods. The number of hidden layers in each MLP is 100. Table 15 shows the architecture details of the actor-critic network for MuJoCo tasks. Table 16 shows the MLP architectures, i.e., GIRIL's encoder and decoder and VAIL's discriminator.

Table 15: Architectures of the actor-critic network for MuJoCo tasks. $\dim(\mathcal{S})$ and $\dim(\mathcal{A})$ represent the state dimension and action dimension, respectively.

| Actor-Critic network | |
|---|---|
| $1 \times \dim(\mathcal{S})$ States | |
| dense $\dim(\mathcal{S}) \to 100$, Tanh | |
| dense $100 \to 100$, Tanh | |
| dense $100 \to \dim(\mathcal{A})$ Gaussian Distribution | dense $100 \to 1$ |
| Actions | Values |

**Hyperparameter settings.** Table 17 summarized the parameter settings for MuJoCo tasks.

Table 16: Architectures of MLP-based GIRIL's *encoder* and *decoder*, and VAIL's discriminator.

| | GIRIL | | VAIL |
|---|---|---|---|
| *encoder* | *decoder* | | discriminator |
| $1 \times \dim(\mathcal{S})$ States and Next State | $1 \times \dim(\mathcal{A})$ Actions and $1 \times \dim(\mathcal{S})$ States | | $1 \times \dim(\mathcal{A})$ Actions and $1 \times \dim(\mathcal{S})$ States |
| dense $\dim(\mathcal{S}) \times 2 \to 100$, Tanh 
 dense $100 \to 100$, Tanh 
 dense $100 \to \boldsymbol{\mu}$, dense $100 \to \boldsymbol{\sigma}$ 
 reparameterization $\to \dim(\mathcal{A})$ | dense $\dim(\mathcal{S})+\dim(\mathcal{A}) \to 100$, Tanh 
 dense $100 \to 100$, Tanh 
 dense $100 \to \dim(\mathcal{S})$ | | dense $\dim(\mathcal{S}) \to 100$, Tanh 
 dense $100 \to 100$, Tanh 
 split 100 into 50, 50 
 dense $50 \to \boldsymbol{\mu}_{E'}$, dense $50 \to \boldsymbol{\sigma}_{E'}$ 
 reparameterization $\to 1$ |
| Latent variables | $\dim(\mathcal{S})$ Predicted next states | | $\boldsymbol{\mu}_{E'}$, $\boldsymbol{\sigma}_{E'}$, discrimination |

Table 17: Hyperparameter settings for MuJoCo tasks.

| Parameter | Setting |
|---|---|
| Actor-critic learning rate | 3e-4, linear decay |
| PPO clipping | 0.1 |
| Reward model learning rate | 3e-4 |
| Discount factor | 0.99 |
| GAE Lambda | 0.95 |
| Critic loss coefficient | 0.5 |
| Entropy regularisation | 0.0 |
| Epochs per batch | 10 |
| Batch size | $32 \times 2048$ (parallel workers $\times$ time steps) |
| Mini batch size | 32 |
| Evaluation frequency | every 6 policy updates |
| GIRIL objective $\alpha$ | 0.01 |
| Mini batch size for training the reward model | 32 |
| Total training epoch of reward model | $10,000$ |
| VAIL's bottleneck loss weight $\beta$ | 1.0 |
| Information constraint $I_c$ | 0.2 |
| $\lambda$ in Algorithm 3 | 10.0 |

**Results.** Figure 2 illustrates the curves of VAIL, GIRIL, and ILDE in the four continuous control tasks. The ablation study results for MuJoCo are illustrated in Figure 3.

## C  PROOF OF THEOREM 4.7

Our proof structure for Theorem 4.7 largely adapts from the proofs in Liu et al. (2023b). However, our analysis different from theirs in three aspects: (1) we consider imitation learning, where the reward is time-varying, (2) we consider MDPs with generalized Eluder dimension, which is slightly more general than Eluder dimension (Russo & Van Roy, 2013) as shown in Zhao et al. (2023), and (3) the analysis leads to a sublinear batch-regret, which is stronger than a sample complexity bound.

First, we have the following extended value difference lemma.

**Lemma C.1** (Value difference lemma). For any policy $\pi$, the value functions $\{V_h^k\}_{h=1}^H$ and $\{Q_h^k\}_{h=1}^H$ returned by $\text{OPE}(\pi, \mathcal{D}, \widetilde{r})$ satisfy the following equation,

$$
V_{1,\pi}^{\widetilde{r}}(s_1) - V_1^k(s_1) = \sum_{h=1}^H \mathbb{E}_{s_h \sim \pi} \left[ \langle Q_h^k(s_h, \cdot), \pi_h(\cdot|s_h) - \pi_h^k(\cdot|s_h) \rangle \right]
$$

$$
- \sum_{h=1}^H \mathbb{E}_{(s_h, a_h) \sim \pi} \left[ Q_h^k(s_h, a_h) - \widetilde{r}(s_h, a_h) - \mathbb{P}_h V_{h+1}^k(s_h, a_h) \right].
$$

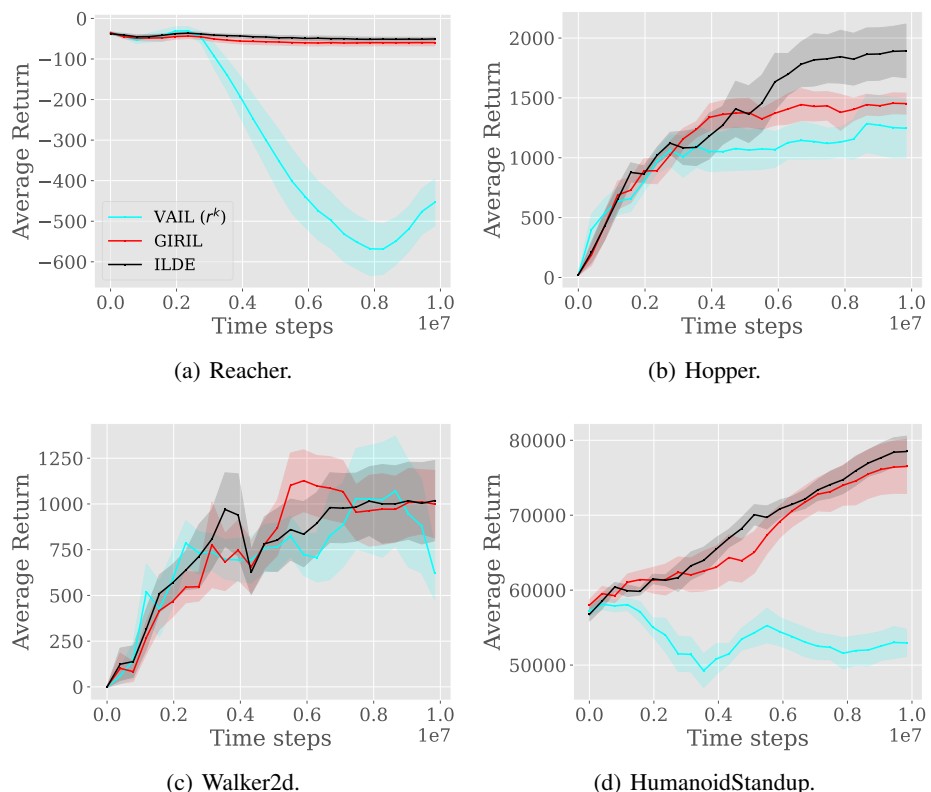

Figure 2: Average return vs. the number of simulation steps on MuJoCo tasks. The solid lines show the mean performance over 10 random seeds. The shaded area represents the standard deviation from the mean.

*Proof.* For any $h \in [H]$, $s \in \mathcal{S}$, we have

$$
\begin{aligned}
V_{h,\pi}^{\widetilde{r}}(s) - V_h^k(s) &= \langle Q_{h,\pi}^{\widetilde{r}}(s,\cdot), \pi_h(\cdot|s)\rangle - \langle Q_h^k(s,\cdot), \pi_h^k(\cdot|s)\rangle \\
&= \langle Q_{h,\pi}^{\widetilde{r}}(s,\cdot) - Q_h^k(s,\cdot), \pi_h(\cdot|s)\rangle - \langle Q_h^k(s,\cdot), \pi_h^k(\cdot|s) - \pi_h(\cdot|s)\rangle \\
&= \langle \mathbb{P}_h V_{h+1,\pi}^{\widetilde{r}}(s,\cdot) - \mathbb{P}_h V_{h+1}^k(s,\cdot), \pi_h(\cdot|s)\rangle - \langle Q_h^k(s,\cdot), \pi_h^k(\cdot|s) - \pi_h(\cdot|s)\rangle \\
&\quad - \langle Q_h^k(s,\cdot) - \widetilde{r}(s,\cdot) - \mathbb{P}_h V_{h+1}^k(s,\cdot), \pi_h(\cdot|s)\rangle, \quad\quad\quad\quad (\text{C.1})
\end{aligned}
$$

where the first equality follows from the definition of the value functions $V_h^k$ and $Q_h^k$, the last equality follows from the Bellman equation for the state-action value function $Q_{h,\pi}$ in (3.2). From (C.1) we further obtain that for any $h \in [H]$, policy $\pi$,

$$
\begin{aligned}
&\mathbb{E}_{s_h \sim \pi}\big[V_{h,\pi}^{\widetilde{r}}(s_h) - V_h^k(s_h)\big] - \mathbb{E}_{s_h \sim \pi}\big[V_{h+1,\pi}^{\widetilde{r}}(s_{h+1}) - V_{h+1}^k(s_{h+1})\big] \\
&= \mathbb{E}_{s_h \sim \pi}\big[\langle Q_h^k(s_h,\cdot), \pi_h(\cdot|s_h) - \pi_h^k(\cdot|s_h)\rangle\big] - \mathbb{E}_{s_h \sim \pi}\big[\langle Q_h^k(s_h,\cdot) - \widetilde{r}(s_h,\cdot) - \mathbb{P}_h V_{h+1}^k(s_h,\cdot), \pi_h(\cdot|s_h)\rangle\big].
\end{aligned}
$$

Taking the sum over $h \in [H]$ yields the following result,

$$
\begin{aligned}
&V_{1,\pi}^{\widetilde{r}}(s_1) - V_1^k(s_1) \\
&= \sum_{h=1}^H \mathbb{E}_{s_h \sim \pi}\big[\langle Q_h^k(s_h,\cdot), \pi_h(\cdot|s_h) - \pi_h^k(\cdot|s_h)\rangle\big] \\
&\quad - \sum_{h=1}^H \mathbb{E}_{s_h \sim \pi}\big[\langle Q_h^k(s_h,\cdot) - \widetilde{r}(s_h,\cdot) - \mathbb{P}_h V_{h+1}^k(s_h,\cdot), \pi_h(\cdot|s_h)\rangle\big] \\
&= \sum_{h=1}^H \mathbb{E}_{s_h \sim \pi}\big[\langle Q_h^k(s_h,\cdot), \pi_h(\cdot|s_h) - \pi_h^k(\cdot|s_h)\rangle\big] - \sum_{h=1}^H \mathbb{E}_{(s_h,a_h) \sim \pi}\big[Q_h^k(s_h,a_h) - \widetilde{r}(s_h,a_h) - \mathbb{P}_h V_{h+1}^k(s_h,a_h)\big].
\end{aligned}
$$

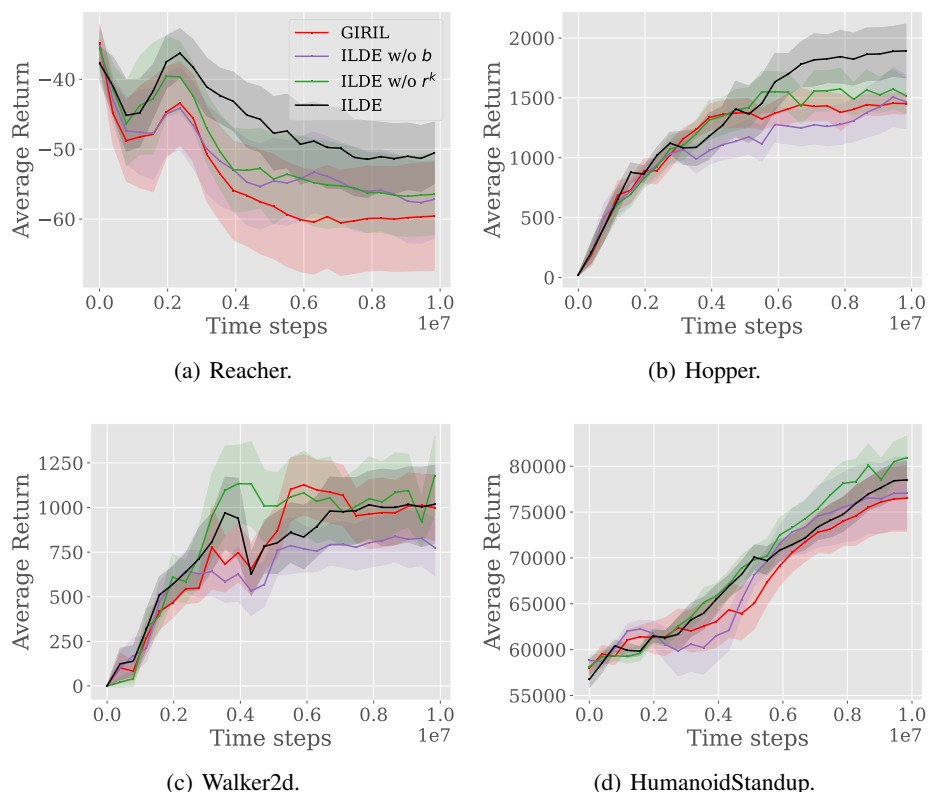

Figure 3: Ablations on MuJoCo tasks. Average return vs. the number of simulation steps on MuJoCo tasks. The solid lines show the mean performance over 10 random seeds. The shaded area represents the standard deviation from the mean.

$\square$

**Lemma C.2** (Lemma 17, Shani et al. 2020). *For any policy $\pi$, $s \in \mathcal{S}$, $h \in [H]$, we have*

$$\sum_{k=1}^{K} \left[ \langle Q_h^k(s, \cdot), \pi_h(\cdot|s) - \pi_h^k(\cdot|s) \rangle \right] \leq \log |\mathcal{A}|/\eta + \eta H^2 K/2.$$

**Lemma C.3.** *Suppose Assumption 4.5 holds. Then with probability at least $1 - \delta$, for all $h \in [H]$, $s \in \mathcal{S}$, $a \in \mathcal{A}$, we have*

$$\left| \widehat{f}_{k,h}(s, a) - \mathbb{P}_h V_{k,h+1}(s, a) \right| \leq \sqrt{8H^2 \log(H \cdot \mathcal{N}_{\mathcal{F}}(\epsilon_{\mathcal{F}})/\delta) + 4\epsilon_{\mathcal{F}} N + \gamma} \cdot D_{\mathcal{F}_h}((s, a); \mathcal{D}_h).$$

*Proof.* We have

$$\sum_{(s_h, a_h) \in \mathcal{D}_h^k} \left( \widehat{f}_{k,h}(s_h, a_h) - \mathbb{P}_h V_{k,h+1}(s_h, a_h) \right)^2$$

$$+ 2 \sum_{(s_h, a_h) \in \mathcal{D}_h^k} \left( \widehat{f}_{k,h}(s_h, a_h) - \mathbb{P}_h V_{k,h+1}(s_h, a_h) \right) \cdot \left( V_{k,h+1}(s_{h+1}) - \mathbb{P}_h V_{k,h+1}(s_h, a_h) \right)$$

$$= \sum_{(s_h, a_h) \in \mathcal{D}_h^k} \left( \widehat{f}_{k,h}(s_h, a_h) - V_{k,h+1}(s_{h+1}) \right)^2 - \sum_{(s_h, a_h) \in \mathcal{D}_h^k} \left( V_{k,h+1}(s_{h+1}) - \mathbb{P}_h V_{k,h+1}(s_h, a_h) \right)^2 \leq 0,$$

where the last inequality follows from the definition of $\widehat{f}_{k,h}$ in Algorithm 2.

Then it follows that for all $h \in [H]$,

$$\sum_{(s_h, a_h) \in \mathcal{D}_h^k} \left(\widehat{f}_{k,h}(s_h, a_h) - \mathbb{P}_h V_{k,h+1}(s_h, a_h)\right)^2$$

$$\leq 2 \sum_{(s_h, a_h) \in \mathcal{D}_h^k} \left(\widehat{f}_{k,h}(s_h, a_h) - \mathbb{P}_h V_{k,h+1}(s_h, a_h)\right) \cdot \left(\mathbb{P}_h V_{k,h+1}(s_h, a_h) - V_{k,h+1}(s_{h+1})\right). \quad (C.2)$$

Note that $V_{k,h+1}$ is a function only depends on data in $\mathcal{D}_{h+1}, \mathcal{D}_{h+2}, \cdots, \mathcal{D}_H$. Hence the right-hand side of the above inequality is a martingale difference sequence.

Consider a function $f \in \mathcal{C}(\mathcal{F}_h, \epsilon_{\mathcal{F}}) \subseteq \mathcal{F}_h$. By Lemma D.2, we have with probability at least $1 - \delta/(H \cdot \mathcal{N}_{\mathcal{F}}(\epsilon_{\mathcal{F}}))$,

$$\sum_{(s_h, a_h) \in \mathcal{D}_h^k} \left(f(s_h, a_h) - \mathbb{P}_h V_{k,h+1}(s_h, a_h)\right) \cdot \left(\mathbb{P}_h V_{k,h+1}(s_h, a_h) - V_{k,h+1}(s_{h+1})\right)$$

$$\leq \frac{1}{4} \sum_{(s_h, a_h) \in \mathcal{D}_h^k} \left(f(s_h, a_h) - \mathbb{P}_h V_{k,h+1}(s_h, a_h)\right)^2 + 2H^2 \log(H \cdot \mathcal{N}_{\mathcal{F}}(\epsilon_{\mathcal{F}})/\delta). \quad (C.3)$$

Combining (C.2) and (C.3) and from the definition of $\epsilon_{\mathcal{F}}$-net $\mathcal{C}(\mathcal{F}_h, \epsilon_{\mathcal{F}})$, we have with probability at least $1 - \delta/H$,

$$\sum_{(s_h, a_h) \in \mathcal{D}_h^k} \left(\widehat{f}_{k,h}(s_h, a_h) - \mathbb{P}_h V_{k,h+1}(s_h, a_h)\right)^2 \leq 8H^2 \log(H \cdot \mathcal{N}_{\mathcal{F}}(\epsilon_{\mathcal{F}})/\delta) + 4\epsilon_{\mathcal{F}} H \cdot N/H.$$

By the definition of $D_{\mathcal{F}_h}^2$, we have with probability at least $1 - \delta/H$,

$$\left|\widehat{f}_{k,h}(s, a) - \mathbb{P}_h V_{k,h+1}(s, a)\right| \leq \sqrt{8H^2 \log(H \cdot \mathcal{N}_{\mathcal{F}}(\epsilon_{\mathcal{F}})/\delta) + 4\epsilon_{\mathcal{F}} N + \gamma} \cdot D_{\mathcal{F}_h}((s, a); \mathcal{D}_h),$$

from which we can complete the proof by applying the union bound over $h \in [H]$. $\qquad \square$

**Lemma C.4.** For any policy $\pi$, suppose we sample $N$ trajectories $\mathcal{D} = \{s_h^{(i)}, a_h^{(i)}\}_{h \in [H], i \in [N]}$. Then, with probability at least $1 - \delta$, we have

$$\mathbb{E}_{z_h \sim \pi} \left[D_{\mathcal{F}_h}(z_h; \mathcal{D})\right] = O\left(\sqrt{\frac{1}{N} \cdot \frac{H^2 + \gamma}{\gamma} \dim_N(\mathcal{F}_h) \log(1/\delta)}\right).$$

*Proof.* It follows from the definition of $\dim_N(\mathcal{F}_h)$ that

$$\dim_N(\mathcal{F}_h) \geq \sum_{i=1}^N \min\left(1, D_{\mathcal{F}_h}^2(z_h^{(i)}; z_h^{[i-1]})\right). \quad (C.4)$$

From Definition 4.3, we further have the following upper bound for cumulative uncertainty,

$$\sum_{i=1}^N D_{\mathcal{F}_h}^2(z_h^{(i)}; z_h^{[i-1]}) \leq \frac{H^2}{\gamma} \sum_{i=1}^N \mathbb{1}\left(D_{\mathcal{F}_h}^2(z_h^{(i)}; z_h^{[i-1]}) > 1\right) + \sum_{i=1}^N \min\left(1, D_{\mathcal{F}_h}^2(z_h^{(i)}; z_h^{[i-1]})\right)$$

$$\leq \left(\frac{H^2}{\gamma} + 1\right) \cdot \dim_N(\mathcal{F}_h). \quad (C.5)$$

Thus,

$$\sum_{i=1}^N D_{\mathcal{F}_h}(z_h^{(i)}; z_h^{[i-1]}) \leq \sqrt{N \cdot \left(\frac{H^2}{\gamma} + 1\right) \cdot \dim_N(\mathcal{F}_h)} \quad (C.6)$$

by AM-GM inequality.

On the other hand, from Azuma-Hoeffding inequality (Lemma D.1), we have with probability at least $1 - \delta$,

$$
\sum_{i=1}^{N} D_{\mathcal{F}_h}(z_h^{(i)}; z_h^{[i-1]}) \geq \sum_{i=1}^{N} \mathbb{E}_{z_h \sim \pi}\left[D_{\mathcal{F}_h}(z_h; z_h^{[i-1]})\right] - O\left(\frac{H}{\sqrt{\gamma}}\sqrt{N \log(1/\delta)}\right)
$$

$$
\geq N \cdot \mathbb{E}_{z_h \sim \pi}\left[D_{\mathcal{F}_h}(z_h; \mathcal{D})\right] - O\left(\frac{H}{\sqrt{\gamma}}\sqrt{N \log(1/\delta)}\right) \tag{C.7}
$$

Substituting (C.6) into (C.7) yields:

$$
\mathbb{E}_{z_h \sim \pi}\left[D_{\mathcal{F}_h}(z_h; \mathcal{D})\right] = O\left(\sqrt{\frac{1}{N} \cdot \frac{H^2 + \gamma}{\gamma} \dim_N(\mathcal{F}_h) \log(1/\delta)}\right). \tag{C.8}
$$

$\square$

**Lemma C.5** (Lemma 3, Liu et al. 2023b)**.** Let $t_k$ be the index of the last policy which is used to collect fresh data at iteration $k$. Suppose we choose $\eta$ and $m$ such that $\eta m \leq 1/H^2$, then for any $k \in \mathbb{N}^+$ and any function $g : \mathcal{S} \times \mathcal{A} \to \mathbb{R}^+$:

$$
\mathbb{E}_{\pi^k}[g(s_h, a_h)] = \Theta\left(\mathbb{E}_{\pi^{t_k}}[g(s_h, a_h)]\right).
$$

**Lemma C.6** (Bounding batched regret $(I_1 + I_3)$)**.** Let $\widetilde{r}_k$ be defined in (7) and update the policy as in Algorithm 1. Suppose that Assumption 4.5 holds. Then with probability at least $1 - 2\delta$,

$$
\sum_{k=1}^{K}\left[J(\pi^*, \widetilde{r}^k) - J(\pi, \widetilde{r}^k)\right] \leq H \log |\mathcal{A}|/\eta + \eta H^3 K/2
$$

$$
+ 2HK \cdot \sqrt{8H^2 \log(H \cdot \mathcal{N}_{\mathcal{F}}(\epsilon_{\mathcal{F}})/\delta) + 4\epsilon_{\mathcal{F}} N + \gamma} \cdot O\left(\sqrt{\frac{H}{N} \cdot \frac{H^2 + \gamma}{\gamma} \dim_N(\mathcal{F}_h) \log(1/\delta)}\right).
$$

*Proof.* Suppose that the high-probability events in Lemma C.3 and Lemma C.5 hold.

$$
\sum_{k=1}^{K}\left[J(\pi^*, \widetilde{r}^k) - J(\pi, \widetilde{r}^k)\right] = \sum_{k=1}^{K}\left(V_{1,\pi^*}^{\widetilde{r}^k}(s_1) - V_1^k(s_1)\right) + \sum_{k=1}^{K}\left(V_1^k(s_1) - V_{1,\pi^k}^{\widetilde{r}^k}(s_1)\right)
$$

$$
\leq \sum_{k=1}^{K}\sum_{h=1}^{H} \mathbb{E}_{s_h \sim \pi^*}\left[\langle Q_h^k(s_h, \cdot), \pi_h(\cdot|s_h) - \pi_h^k(\cdot|s_h)\rangle\right]
$$

$$
+ \sum_{k=1}^{K}\sum_{h=1}^{H} \mathbb{E}_{(s_h, a_h) \sim \pi^k}\left[Q_h^k(s_h, a_h) - \widetilde{r}(s_h, a_h) - \mathbb{P}_h V_{h+1}^k(s_h, a_h)\right]
$$

$$
\leq H \log |\mathcal{A}|/\eta + \eta H^3 K/2 + 2HK \cdot \sqrt{8H^2 \log(H \cdot \mathcal{N}_{\mathcal{F}}(\epsilon_{\mathcal{F}})/\delta) + 4\epsilon_{\mathcal{F}} N + \gamma}
$$

$$
\cdot O\left(\sqrt{\frac{H}{N} \cdot \frac{H^2 + \gamma}{\gamma} \dim_N(\mathcal{F}) \log(1/\delta)}\right)
$$

where the first inequality is obtained by applying the value difference lemma C.1 to both terms and then applying Lemma C.3. $\square$

**Lemma C.7** (Upper bound for $I_2$)**.** If we choose $\eta_{\boldsymbol{\theta}} = O(1/\sqrt{H^2 K})$ in Algorithm 1 and update the reward function as shown in (4.3), then with probability $1 - \delta$,

$$
\sum_{k=1}^{K}\left[L(\pi^k, r) - L(\pi^k, r^k)\right] \leq \widetilde{O}\left(\sqrt{H^2 K} + \epsilon_E K\right)
$$

*Proof.* From Lemma D.3, we have for any $r \in \mathcal{R}$,

$$
\sum_{k=1}^{K}\left[\widehat{L}(\pi^k, r) - \widehat{L}(\pi^k, r^k)\right] \leq O(1/\eta_{\boldsymbol{\theta}} + \eta_{\boldsymbol{\theta}} H^2 K/2). \tag{C.9}
$$

From Lemmas D.1 and D.4, with probability at least $1 - \delta$,

$$\sum_{k=1}^{K} \left[ L(\pi^k, r) - L(\pi^k, r^k) \right]$$

$$\leq O(1/\eta_{\boldsymbol{\theta}} + \eta_{\boldsymbol{\theta}} H^2 K / 2) + O\left( \sqrt{H^2 K \log(1/\delta)} \right) + O(\epsilon_E K), \qquad \text{(C.10)}$$

from which we can complete the proof by substituting the value of $\eta_{\boldsymbol{\theta}}$ into (C.10). $\qquad\square$

**Theorem C.8** (Restatement of Theorem 4.7). Suppose Assumptions 4.5, 4.1 and 4.2 hold. If we set $\gamma = H^2$, $\epsilon_{\mathcal{F}} = 1/N$, $\eta = \sqrt{\log |\mathcal{A}|}/H\sqrt{K}$, $m = \lfloor \sqrt{K}/H\sqrt{\log |\mathcal{A}|} \rfloor$, $\eta_{\boldsymbol{\theta}} = O(1/\sqrt{H^2 K})$, and $N = \lceil KH \dim_T(\mathcal{F}) \log \mathcal{N}_{\epsilon_{\mathcal{F}}}(\mathcal{F}) / \sqrt{\log |\mathcal{A}|} \rceil$, where $K = \left\lceil \left( \frac{T}{H^2 \dim_T(\mathcal{F}) \log \mathcal{N}_{\epsilon_{\mathcal{F}}}(\mathcal{F})} \right)^{2/3} \right\rceil$, then with probability at least $1 - \delta$, Algorithm 1 yields a regret of

$$\widetilde{O} \left( H^{8/3} \big( \dim_T(\mathcal{F}) \log \mathcal{N}_{\epsilon_{\mathcal{F}}}(\mathcal{F}) \big)^{1/3} T^{2/3} + \epsilon_E T \right).$$

*Proof for Theorem 4.7.* Throughout the proof, we suppose the events in Lemma C.6 and Lemma C.7 hold simultaneously.

From the definition of Regret,

$$\text{Regret}(T) \leq (N/m) \cdot \max_{r \in \mathcal{R}} \sum_{k=1}^{K} \mathbb{E}[\ell(\pi^{t_k}, r)] - \ell(\pi^*, r)$$

$$\leq (N/m) \cdot O \left( \max_{r \in \mathcal{R}} \sum_{k=1}^{K} \ell(\pi^k, r) - \ell(\pi^*, r) \right), \qquad \text{(C.11)}$$

where the last inequality follows from the definition of $t_k$.

Then we consider the following decomposition for the regret term,

$$\max_{r \in \mathcal{R}} \sum_{k=1}^{K} \ell(\pi^k, r) - \ell(\pi^*, r) \leq \underbrace{\sum_{k=1}^{K} \left[ J(\pi^*, r^k) - J(\pi^k, r^k) \right]}_{I_1}$$

$$+ \underbrace{\sup_{r \in \mathcal{R}} \sum_{k=1}^{K} \left[ L(\pi^k, r) - L(\pi^k, r^k) \right]}_{I_2} + \lambda \underbrace{\sum_{k=1}^{K} \left[ \text{Int}(\pi^*; \tau^E) - \text{Int}(\pi^k; \tau^E) \right]}_{I_3}, \qquad \text{(C.12)}$$

where $I_1 + I_3$ can be controlled by the classic analysis for optimistic policy optimization, $I_2$ is the regret of an online learning algorithm for the reward function.

From Lemma C.6,

$$I_1 + I_3 \leq \widetilde{O}(H \log |\mathcal{A}|/\eta + \eta H^3 K/2) + \widetilde{O} \left( H^2 K \sqrt{\frac{H}{N} \log(\mathcal{N}_{\mathcal{F}}(\epsilon_{\mathcal{F}}) \dim_T(\mathcal{F}))} \right)$$

$$\leq \widetilde{O} \left( H^2 \sqrt{K \log |\mathcal{A}|} \right), \qquad \text{(C.13)}$$

where the last inequality holds due to the value of $\eta$, $N$ we are choosing.

From Lemma C.7,

$$I_2 \leq \widetilde{O} \left( \sqrt{H^2 K} + \epsilon_E K \right). \qquad \text{(C.14)}$$

Substituting (C.13) and (C.14) into (C.11),

$$\max_{r \in \mathcal{R}} \sum_{k=1}^{K} \ell(\pi^k, r) - \ell(\pi^*, r) \leq \widetilde{O} \left( H^2 \sqrt{K \log |\mathcal{A}|} + \epsilon_E K \right).$$

After we further substitute the value of hyperparameters into (C.11), we obtain

$$\text{Regret}(T) \leq \widetilde{O}\left(H^{8/3}\left(\dim_T(\mathcal{F}) \log \mathcal{N}_{\epsilon_{\mathcal{F}}}(\mathcal{F})\right)^{1/3} T^{2/3} + \epsilon_E T\right).$$

$\square$

# D  AUXILIARY LEMMAS

**Lemma D.1** (Azuma-Hoeffding inequality). Let $\{x_i\}_{i=1}^n$ be a martingale difference sequence with respect to a filtration $\{\mathcal{G}_i\}$ satisfying $|x_i| \leq M$ for some constant $M$, $x_i$ is $\mathcal{G}_{i+1}$-measurable, $\mathbb{E}[x_i|\mathcal{G}_i] = 0$. Then for any $0 < \delta < 1$, with probability at least $1 - \delta$, we have

$$\sum_{i=1}^n x_i \leq M\sqrt{2n\log(1/\delta)}.$$

**Lemma D.2** (Self-normalized bound for scalar-valued martingales). Consider random variables $(v_n|n \in \mathbb{N})$ adapted to the filtration $(\mathcal{H}_n : n = 0, 1, ...)$. Let $\{\eta_i\}_{i=1}^\infty$ be a sequence of real-valued random variables which is $\mathcal{H}_{i+1}$-measurable and is conditionally $\sigma$-sub-Gaussian. Then for an arbitrarily chosen $\lambda > 0$, for any $\delta > 0$, with probability at least $1 - \delta$, it holds that

$$\sum_{i=1}^n \epsilon_i v_i \leq \frac{\lambda \sigma^2}{2} \cdot \sum_{i=1}^n v_i^2 + \log(1/\delta)/\lambda \qquad \forall n \in \mathbb{N}.$$

**Lemma D.3** (Regret bound of online gradient descent, Orabona 2019). Let $V \subseteq \mathbb{R}^d$ a non-empty closed convex set with diameter $D$, i.e., $\max_{\boldsymbol{x},\boldsymbol{y} \in V} \|\boldsymbol{x} - \boldsymbol{y}\|_2 \leq D$. Let $\ell_1, \cdots, \ell_T$ an arbitrary sequence of convex functions $\ell_t : V \to \mathbb{R}$ differentiable in open sets containing $V$. Pick any $\boldsymbol{x}_1 \in V$ and assume $\eta_{t+1} \leq \eta_t$, $t = 1, \ldots, T$. Then, $\forall \boldsymbol{u} \in V$, the following regret bound holds:

$$\sum_{t=1}^T (\ell_t(\boldsymbol{x}_t) - \ell_t(\boldsymbol{u})) \leq \frac{D^2}{2\eta_T} + \sum_{t=1}^T \frac{\eta_t}{2} \|\boldsymbol{g}_t\|_2^2.$$

Moreover, if $\eta_t$ is constant, i.e., $\eta_t = \eta \, \forall t = 1, \cdots, T$, we have

$$\sum_{t=1}^T (\ell_t(\boldsymbol{x}_t) - \ell_t(\boldsymbol{u})) \leq \frac{\|\boldsymbol{u} - \boldsymbol{x}_1\|_2^2}{2\eta} + \frac{\eta}{2} \sum_{t=1}^T \|\boldsymbol{g}_t\|_2^2.$$

**Lemma D.4** (Lemma 9, Liu et al. 2023b). There exists an absolute constant $c > 0$ such that for any pair of policies $\pi, \widehat{\pi}$ satisfying $\widehat{\pi}_h(a \mid s) \propto \pi_h(a \mid s) \times \exp(L_h(s,a))$, where $\{L_h(s,a)\}_{h \in [H]}$ is a set of functions from $\mathcal{S} \times \mathcal{A}$ to $[-1/H, 1/H]$, it holds that for any $\tau_H := (s_1, a_1, \ldots, s_H, a_H) \in (\mathcal{S} \times \mathcal{A})^H$:

$$\mathbb{P}^{\widehat{\pi}}(\tau_H) \leq c \times \mathbb{P}^\pi(\tau_H).$$

