# OpenReview forum: "Beyond-Expert Performance with Limited Demonstrations: Efficient Imitation Learning with Double Exploration"
_ICLR.cc/2025/Conference — ICLR 2025 Poster_

### Official Review · Reviewer_4bF6 · 2024-10-22

**Soundness:** 3
**Presentation:** 4
**Contribution:** 3
**Rating:** 8
**Confidence:** 4

**Summary:**

This paper proposes a new framework for imitation learning, which succinctly, leverage informed interaction to improve policy performance in the absence of an abundance of expert demonstrations. In particular, two exploration strategies are considered: optimistic policy optimization with an exploration bonus, and curiosity-driven exploration of states beyond the demonstration trajectories. The authors provide extensive theoretical justification and prove performance bounds to support their algorithm, and also provide empirical results to support their method, testing metrics in both sample efficiency and final performance.

**Strengths:**

* The proposed method seems novel and to be well-grounded in theoretical derivations. While learning from limited expert demonstrations has been explored in prior work, this work introduces a unique combination of two exploration strategies to tackle this regime.
* The paper provides solid theoretical foundations for ILDR, and is thorough in stating their assumptions. As noted, Theorem 4.7 is the first theoretical guarantee for imitation learning in MDPs with nonlinear
function approximation.
* The problem is well motivated, and algorithms are well presented. Particularly, the integration of optimistic policy optimization and curiosity-driven exploration are clearly labeled. The empirical experiments also do a good job ablating on these two exploration models, highlighting both are needed for best performance.
* Relatedly, the empirical results are promising, demonstrating that ILDE achieves better performance over baselines on metrics such as final reward, # games better than expert, and average performance against experts over the considered environments.

**Weaknesses:**

* There are more modern baselines that come to mind when looking at the few expert-demonstration regime, such as IQLearn [1], which also demonstrate reasonable performance in the low expert-data regime, or the model-free and/or model-based inverse RL algorithms in Hybrid Inverse Reinforcement Learning [2], which also prove competitive sample complexity bounds. The experiments section could benefit from slightly more baselines.
* Although the theoretical framework is well-explained, the practical implementation details could use a bit more clarity. In particular, how are exploration bonuses computed in practice, and how much work is involved in hyperparameter search (i.e. the original ranges considered, type of search used, and how the authors recommend searching over these hyperparameters when faced with a new environment/dataset)? (Relatedly, see Question 2 below.)
* Relatedly, I personally felt the tradeoff between exploration and imitation could have been explored a bit more thoroughly in the paper. For example, I'm curious how algorithms may perform in situations where the demonstrations may not be fully optimal. (Relatedly, see Question 3 below.)
* The ILDR-NPG variant as I understood it seems susceptible to incur large computational costs due to using natural policy gradients for policy optimization. The paper doesn't seem to fully address this issue/potential ways to mitigate it.


[1] Garg, Divyansh, et al. "Iq-learn: Inverse soft-q learning for imitation." Advances in Neural Information Processing Systems 34 (2021): 4028-4039.

[2] Ren, Juntao, et al. "Hybrid inverse reinforcement learning." International Conference on Machine Learning (2024).

**Questions:**

1. While the authors provide a practical instantiation, I am under the impression that proposed ILDE framework should be flexible, and can be generalized to other empirical instantiations. If this is true, have the authors explored applying these exploration bonuses on top of existing methods and seeing how they fair?
2. How sensitive is ILDE to the choice of hyperparameters, such as the exploration bonus weight and the learning rate for curiosity-driven exploration? What is the amount of hyperparameter tuning that one had to search over in order to reach the optimal performance for these exploration bonus trainings?
3. How does ILDE perform in environments where the expert demonstrations are noisy or imperfect? What is the consequence of data quality on the two exploration bonuses, and how much additional tuning may be needed to stabilize performance? This would provide insight into ILDE’s robustness in real-world settings.
4. I'm wondering whether ILDE could be combined with techniques in model-based (inverse) reinforcement learning to further improve sample efficiency?

---

> ### Author Response · Authors · 2024-11-24
> **Response to reviewer  4bF6 (Part I)**
>
> We thank the reviewer for the many positive comments about our theoretical derivations, uniqueness and novelty of the framework, presentation, and empirical evaluation.
>
> **Q1.** There are more modern baselines that come to mind when looking at the few expert-demonstration regime, such as IQLearn [1], which also demonstrate reasonable performance in the low expert-data regime, or the model-free and/or model-based inverse RL algorithms in Hybrid Inverse Reinforcement Learning [2], which also prove competitive sample complexity bounds. The experiments section could benefit from slightly more baselines.
>
> **A1.** Thanks for your suggestions. We have included IQ-Learn on our Atari task setting (with 10% of a one-life demonstration). The results below show that IQ-Learn does not perform well on Atari tasks, since the demonstration data is extremely limited. We didn’t include HyPE in the Atari task because their code doesn’t support the Atari environment.
> |Atari Games | IQ-Learn|
> | -------- | -------- |
> |BeamRider | 0.0$\pm$0.0  |
> |DemonAttack |  88.5$\pm$92  |
> |BattleZone | 3,800$\pm$870   |
> |Qbert |   225$\pm$125 |
> |Krull |   4,500$\pm$2,030  |
> |StarGunner |  200$\pm$100  |
>
>
> Instead, we implement ILDE on top of the suggested adversarial IL method HyPE, and named it as **HyPE-ILDE**. We test HyPE and HyPE-ILDE on the MuJoCo tasks with **one demonstration**, instead of using 64 demonstrations as in HyPE’s paper. The results show that ILDE clearly improves the performance of HyPE even on the low-data regime:
> |MuJoCo Tasks | HyPE | HyPE-ILDE|
> | -------- | -------- | -------- |
> | Ant-v3 | 785.8$\pm$528.6 | **813.8$\pm$833.0** |
> | Hopper-v3 | 2995.0$\pm$745.4 | **3372.6$\pm$359.4** |
>
> ---
>
> **Q2.** Although the theoretical framework is well-explained, the practical implementation details could use a bit more clarity. In particular, how are exploration bonuses computed in practice?
>
> **A2.** How exploration bonuses are computed in practice: the bonus b is the state entropy, implemented using the formula we give with the kNN being computed over each batch. The L is implemented according to GIRIL, i.e. using a conditional VAE. We refer the reviewer to Appendix B.2 and in particular to B.2.2 and B.2.3.
>
> ---
>
> **Q3.** How much work is involved in hyperparameter search (i.e. the original ranges considered, type of search used, and how the authors recommend searching over these hyperparameters when faced with a new environment/dataset)? (Relatedly, see Question 2 below.)
>
> **A3.** We used the same hyperparameter setting for the different experiments within a domain.
> It was based on deriving a scaling factor from the observed curiosity scores.
>
> ---
>
> **Q4.** Relatedly, I personally felt the tradeoff between exploration and imitation could have been explored a bit more thoroughly in the paper. For example, I'm curious how algorithms may perform in situations where the demonstrations may not be fully optimal. (Relatedly, see Question 3 below.)
>
> **A4.** The present setting is already suboptimal in the following sense. First, the expert data are obtained from sub-optimal demonstrators obtained from PPO runs. Second, the demonstration data are very sparse, having just 10% of a one-life demonstration in the Atari games and just one episode in the MuJoCo experiments.

---

> ### Author Response · Authors · 2024-11-24
> **Response to reviewer 4bF6 (Part II)**
>
> **Q5.** The ILDR-NPG variant as I understood it seems susceptible to incur large computational costs due to using natural policy gradients for policy optimization. The paper doesn't seem to fully address this issue/potential ways to mitigate it.
>
> **A5.** You are correct that the NPG-based approach is computationally intensive. We primarily use this variant for theoretical demonstration purposes. The reason is that in the value iteration subroutine, we need to visit all state-action pairs to achieve a relatively accurate estimation of the $-function, which is critical for the theoretical analysis of policy-based RL algorithms. In practical applications, however, this subroutine is not necessary. Instead, we leverage the generalization capabilities of neural networks to estimate the value function for each state-action pair, as outlined in the practical version of our method. This is an open problem how to analyze practical policy optimization methods like PPO.
>
> ---
>
> [1] Garg, Divyansh, et al. "Iq-learn: Inverse soft-q learning for imitation." Advances in Neural Information Processing Systems 34 (2021): 4028-4039.
> [2] Ren, Juntao, et al. "Hybrid inverse reinforcement learning." International Conference on Machine Learning (2024).
>
> ---
>
> **Q6.** While the authors provide a practical instantiation, I am under the impression that proposed ILDE framework should be flexible, and can be generalized to other empirical instantiations. If this is true, have the authors explored applying these exploration bonuses on top of existing methods and seeing how they fair?
>
> **A6.** The authors have now also implemented HyPE as an alternative adversarial IL base-learner within our ILDE framework. We implement ILDE on top of the suggested adversarial imitation learning method HyPE and test on the MuJoCo tasks with **one demonstration**. Note that the HyPE uses 64 demonstrations in their paper. The results show that ILDE clearly improves the performance of HyPE even with the number of demonstrations as limited as one:
> |MuJoCo Tasks | HyPE | HyPE-ILDE|
> | -------- | -------- | -------- |
> | Ant-v3 | 785.8$\pm$528.6 | **813.8$\pm$833.0** |
> | Hopper-v3 | 2995.0$\pm$745.4 | **3372.6$\pm$359.4** |
>
> **Q7.** How sensitive is ILDE to the choice of hyperparameters, such as the exploration bonus weight and the learning rate for curiosity-driven exploration? What is the amount of hyperparameter tuning that one had to search over in order to reach the optimal performance for these exploration bonus trainings?
>
> **A7.** We fixed these quantities without tuning. However, to further explore this question, we have changed the lambda parameter from 20, to 10, 5 and 2. Results show the ILDE’s performance slightly increases with the decreasing of $\lambda$.
> | $\lambda$ | 20 |  10 | 5 | 2 |
> | -------- | -------- | -------- | -------- | -------- |
> |BeamRider | 8,973$\pm$1,315  |  11,453$\pm$2,319  |  11,947$\pm$1,228  | 12,600$\pm$1,351 |
>
> ---
>
> **Q8.** How does ILDE perform in environments where the expert demonstrations are noisy or imperfect? What is the consequence of data quality on the two exploration bonuses, and how much additional tuning may be needed to stabilize performance? This would provide insight into ILDE’s robustness in real-world settings.
>
> **A8.** To test the robustness of ILDE, we generate noisy demonstrations by injecting noise into the environment in the following manner: with probability $p_{tremble}$, a random action is executed instead of the one chosen by the policy [2]. Due to the limited time and computational resources, we test the robustness of ILDE for noisy demonstrations on BeamRider without tuning any hyperparameters. Results show that ILDE consistently outperforms the expert (1,918$\pm$645) and GIRIL (8,524$\pm$1,085) on the noisy demonstrations with noise rate ($p_{tremble}$) of 0.01, 0.05, 0.1, 0.3 and 0.5, providing the insight into ILDE’s robustness in real-world settings.
>
> |Noise rate ($p_{tremble}$) | 0.0 |  0.01 | 0.05 | 0.1 | 0.3 | 0.5 |
> | -------- | -------- | -------- | -------- | -------- | -------- | -------- |
> |BeamRider | 11,453$\pm$2,319 |  12,353$\pm$2,753 |  8,893$\pm$951 | 9,935$\pm$1,952  | 9,728$\pm$1,847 | 10,735$\pm$1,198 |
>
> ---
>
> **Q9.** I'm wondering whether ILDE could be combined with techniques in model-based (inverse) reinforcement learning to further improve sample efficiency?
>
> **A9.** As show in *A6*, we have demonstrated that ILDE can improve the performance of the model-free Inverse reinforcement learning method HyPE on the low data regime. Since its model-based counterpart HyPER has shown much better sample efficiency than HyPE. Therefore, we believe that HyPER with ILDE can also have much better sample efficiency than HyPE and HyPE-ILDE.

---

> > ### Comment · Reviewer_4bF6 · 2024-11-25
> >
> > I thank their reviewers for their diligence in following-up with experiments. I would strongly urge reviewers to find a place to include them in the main paper, as I believe they are insightful to possible questions. I will maintain my original rating.

---

> > > ### Author Response · Authors · 2024-11-25
> > >
> > > Thank you very much for your strong support and helpful advice! We will include the new experimental results and discussions in our revision.

---

### Official Review · Reviewer_FEJp · 2024-10-29

**Soundness:** 1
**Presentation:** 2
**Contribution:** 2
**Rating:** 3
**Confidence:** 4

**Summary:**

This paper introduces a novel imitation learning algorithm, ILDE (Imitation Learning with Double Exploration). The goal of ILDE is to achieve beyond-expert performance by improving sample efficiency and addressing the challenges posed by limited demonstrations. ILDE implements two exploration strategies: (1) optimistic policy optimization with an exploration bonus that encourages the exploration of high-uncertainty state-action pairs, and (2) curiosity-driven exploration that rewards states deviating from the expert’s demonstration trajectory. Empirical results show that ILDE outperforms state-of-the-art algorithms in terms of sample efficiency and return on Atari and MuJoCo tasks. The authors also provide a theoretical framework, showing that ILDE’s regret grows sublinearly with the number of episodes.

**Strengths:**

1. This paper proposes a novel ILDE framework, which combines optimistic exploration with curiosity-driven exploration for imitation learning.

**Weaknesses:**

1. **Theoretical Limitations**
    1. First, the assumptions made in this paper are overly strong. In Assumption 4.2, the authors directly assume that the statistical error resulting from using a finite set of expert demonstrations is small. This assumption is both strong and unreasonable, as a primary goal in imitation learning theory is to examine the relationship between the number of expert trajectories and the error bound [1]. Additionally, Assumption 4.1 presumes that the loss function is convex with respect to the reward parameters, making the reward optimization a straightforward online convex optimization problem. In practice, however, the reward optimization problem may be highly non-convex due to the use of neural networks.
    2. Second, the proposed method in Algorithm 1 requires knowledge of the entire MDP transition function. In line 7 of Algorithm 2, computing the intrinsic reward $\mathcal{L}_{\tau^{E}, h}$ depends on the true transition function. However, if the true transition function were known, online exploration would no longer be necessary.
2. **Experimental Limitations**

    In the experiments, this paper does not compare its approach with current state-of-the-art (SOTA) imitation learning (IL) methods. In Atari games, the SOTA method is IQ-Learn [2]. In MuJoCo environments, the SOTA methods include IQ-Learn and HyPE [3]. The absence of these methods in the experiments makes it unclear whether the proposed approach outperforms current methods.

3. **Unsubstantiated Conjecture**

    The goal of this paper is to develop an IL approach capable of achieving performance beyond the expert level. To this end, the paper is based on the conjecture that optimizing the uncertainty-regularized loss shown in Eq. (3.6) enables performance surpassing the expert’s. However, no evidence or justification is provided to support this conjecture.


References:

[1] Nived Rajaraman et al., Toward the fundamental limits of imitation learning.

[2] Divyansh Garg et al., Iq-learn: Inverse soft-q learning for imitation.

[3] Juntao Ren et al., Hybrid inverse reinforcement learning.

**Questions:**

Typos:

1. In line 41, the reference of GAIL is wrong.
2. In line 256, “desccent” should be descent.

---

> ### Author Response · Authors · 2024-11-23
> **Response to Reviewer FEJp (Part I)**
>
> Thanks for your insightful comments. While we are working on the experiments of IQ-Learn and HyPE, we address your other concerns point-by-point as follows.
>
> **Q1.** The assumptions made in this paper are overly strong. In Assumption 4.2, the authors directly assume that the statistical error resulting from using a finite set of expert demonstrations is small. This assumption is both strong and unreasonable, as a primary goal in imitation learning theory is to examine the relationship between the number of expert trajectories and the error bound [1]. Additionally, Assumption 4.1 presumes that the loss function is convex with respect to the reward parameters, making the reward optimization a straightforward online convex optimization problem. In practice, however, the reward optimization problem may be highly non-convex due to the use of neural networks.
>
> **A1.** We respectfully ***disagree*** with this statement.
> Assumption 4.2 is indeed a weak assumption in which $\epsilon_E$ can be bounded with high probability in the worst case. Here is the proof:
> Let $N_\mathcal{R}(\epsilon)$ be the $\epsilon$-covering number of reward function class with respect to infinity norm. Using union bound over all the reward functions in the reward function class and Azuma-Hoeffding Inequality, it holds with probability $1 - \delta$ that,
> $$|\frac{1}{n} \sum_{i = 1}^n \sum_{h = 1}^H r(s_h^{(j)}, a_h^{(j)}) - J(\pi^E, r)| \le H \sqrt{(2 / n) \log(2 N_\mathcal{R}(\epsilon) / \delta)}+ \epsilon \cdot H. $$
>
> The assumption we introduce here is actually making our results **general** so that when the quality of the trajectories is high enough to achieve lower $\epsilon_E$, the sample complexity will be better. This could also explain why the learner can learn good policy with limited trajectories in many experiments.
>
> As for Assumption 4.1, Convexity and Lipschitz Continuity are very standard assumption in the literature of optimization. With these assumptions, we simplify the theoretical analysis of imitation learning frameworks by ensuring that the optimization problems encountered are well-posed and tractable. Convexity guarantees that the problem has a unique global optimum, while Lipschitz continuity ensures controlled sensitivity to variations in the input. These standard assumptions allow us to focus on the core challenges of imitation learning without being encumbered by pathological cases, aligning with established practices in the optimization literature. It is an open question how to design a provable online learning algorithm for loss of a neural network, and we think it should be a challenging future work of independent interest. In the theoretical understanding of imitation learning, we already generalize the previous assumption considered in [4], where all the reward functions are assumed to be linear functions.
>
> ---
>
> **Q2.** The proposed method in Algorithm 1 requires knowledge of the entire MDP transition function. In line 7 of Algorithm 2, computing the intrinsic reward depends on the true transition function. However, if the true transition function were known, online exploration would no longer be necessary.
>
> **A2.** The authors believe that line 4 of Algorithm 2 may be overlooked. The transition dynamics are not known and we use a value targeted regression approach to approximate the state-action value function. Through this step the transition probability is essentially estimated from the trajectory collected by the agent.
>
> ---

---

> ### Author Response · Authors · 2024-11-23
> **Response to Reviewer FEJp (Part II)**
>
> **Q4.** The goal of this paper is to develop an IL approach capable of achieving performance beyond the expert level. To this end, the paper is based on the conjecture that optimizing the uncertainty-regularized loss shown in Eq. (3.6) enables performance surpassing the expert’s. However, no evidence or justification is provided to support this conjecture.
>
> **A4.** As shown in [1], by optimizing the classic objective one could not expect to achieve beyond-expert performance even with a large amount of demonstration trajectories. Thus, it is natural to consider self-supervised RL frameworks which does not reply on explicit reward but makes use of the potential correlation between the transition dynamics and reward function.
> In [6], a large of scale of experiments have shown that dynamics-based curiosity is a powerful self-supervised exploration strategy in a variety of environments. In their paper, it is discussed that **empirical RL environments typically require the learner to learn non-trivial skills which coincides with the curiosity-reward signal that encourages the learner to explore hard-to-visit states**. Intrinsic reward proposed by [5] is essentially a variant of curiosity reward which uses VAE model to better characterize the uncertainty of a state-action pair. Previous works [6, 5] have provided **sufficient** empirical evidence of the effectiveness of the intrinsic reward and our experiments show that this is indeed the case. The main contribution of our paper is to provide a novel and principled way to solve Eq. 3.6. We note that there is limited theoretical understanding of curiosity reward at this point, and we think this is a challenging but important avenue for future works. For readers to better understand this intuition, we would also include this discussion in the revision.
>
> **Q5.** Typos:
> In line 41, the reference of GAIL is wrong.
> In line 256, “desccent” should be descent.
>
> **A5.** Thank you for pointing them out! We have corrected both typos.
>
> [1] Nived Rajaraman et al., Toward the fundamental limits of imitation learning, NIPS 2020.
>
> [2] Divyansh Garg et al., Iq-learn: Inverse soft-q learning for imitation.
>
> [3] Juntao Ren et al., Hybrid inverse reinforcement learning.
>
> [4] Liu et al., Learning from demonstration: Provably efficient adversarial policy imitation with linear function approximation, ICML 2022.
>
> [5] Yu et al., Intrinsic Reward Driven Imitation Learning via Generative Model, ICML 2020.
>
> [6] Burda et al., Large-Scale Study of Curiosity-Driven Learning, ICLR 2019.

---

> ### Author Response · Authors · 2024-11-24
> **Reponse to Reviewer FEJp (Part III)**
>
> **Q3.** In the experiments, this paper does not compare its approach with current state-of-the-art (SOTA) imitation learning (IL) methods. In Atari games, the SOTA method is IQ-Learn [2]. In MuJoCo environments, the SOTA methods include IQ-Learn and HyPE [3]. The absence of these methods in the experiments makes it unclear whether the proposed approach outperforms current method.
>
> **A3.** Thanks for your suggestions. We have included IQ-Learn on our Atari task setting (with 10% of a one-life demonstration). The results below show that IQ-Learn does not perform well on Atari tasks, since the demonstration data is extremely limited. We didn’t include HyPE in the Atari task because their code doesn’t support the Atari environment.
> |Atari Games | IQ-Learn|
> | -------- | -------- |
> |BeamRider | 0.0$\pm$0.0  |
> |DemonAttack |  88.5$\pm$92  |
> |BattleZone | 3,800$\pm$870   |
> |Qbert |   225$\pm$125 |
> |Krull |   4,500$\pm$2,030  |
> |StarGunner |  200$\pm$100  |
>
>
> Instead, we implement ILDE on top of the suggested adversarial IL method HyPE, and named it as **HyPE-ILDE**. We test HyPE and HyPE-ILDE on the MuJoCo tasks with **one demonstration**, instead of using 64 demonstrations as in HyPE’s paper. The results show that ILDE clearly improves the performance of HyPE even on the low-data regime:
> |MuJoCo Tasks | HyPE | HyPE-ILDE|
> | -------- | -------- | -------- |
> | Ant-v3 | 785.8$\pm$528.6 | **813.8$\pm$833.0** |
> | Hopper-v3 | 2995.0$\pm$745.4 | **3372.6$\pm$359.4** |

---

> ### Author Response · Authors · 2024-11-24
> **Follow up with Reviewer FEJp**
>
> Thank you again for your helpful feedback! We sincerely hope that we have adequately addressed your questions and concerns. Specifically,
>
> * We justified all the assumptions we are using in our theoretical analysis.
>
> * We provided detailed explanations on the motivation of our optimization objective and showed the evidence to support our conjecture.
>
> * We incorporated additional baselines into our experiments. Our results indicated that ILDE still outperformed all the other baselines.
>
> We hope these responses address your questions adequately. We would like to follow up and see if there are remaining questions about our rebuttal, for which we are happy to provide further clarifications. Thank you for your time and efforts in reviewing this paper!
>
> Best, \
> authors

---

> > ### Comment · Reviewer_FEJp · 2024-11-27
> > **Thanks for the detailed response!**
> >
> > **Regarding Assumption 4.1.**
> >
> > I understand that convexity and Lipschitz continuity are standard assumptions in optimization. However, the paper emphasizes that one of its contributions is providing the first theoretical guarantee for imitation learning (IL) with general function approximation. In the context of general function approximation, assuming the loss is convex and Lipschitz continuous significantly excludes many common function classes, such as neural networks, which weakens the contribution to general function approximation.
> >
> > **Regarding the assumption on knowing the transition function.**
> >
> > In line 7 of Algorithm 2, we need to calculate the intrinsic reward $\mathcal{L}\_{\tau^E, h} (s, a)$, which is defined as $\mathcal{L}\_{\tau^E, h} (s, a)= \mathbb{E}_{\widehat{s}^{\prime} \sim \widehat{\mathbb{P}}_h(\cdot \mid s, a), s^{\prime} \sim \mathbb{P}_h(\cdot \mid s, a)}\left[\mathcal{L}\left(\widehat{s}^{\prime}, s^{\prime}\right)\right]$. Calculating this expectation requires knowing the transition function.
> >
> > **Regarding the assumption that the target task to imitate is aligned with seeking novelty.**
> >
> > As discussed by Reviewer 4joU, this paper relies on the assumption that the target task to imitate is aligned with seeking novelty. First, this assumption does not hold in many practical tasks where seeking novelty typically raises safety concerns. Second, the paper does not provide a formal formulation of this assumption. As a result, it remains unclear whether the proposed algorithm can, in principle, achieve performance beyond the expert level.

---

> ### Author Response · Authors · 2024-11-28
> **Response to Reviewer FEJp**
>
> Thank you very much for your further feedback. We would like to address your further concerns point-by-point.
>
> * **Regarding the assumption on knowing the transition function:**
>
>     In our experiments, all the Atari games and MuJoCo tasks are environments with deterministic transition dynamics. In these environments, we can empirically compute the intrinsic reward $\mathcal{L}\_{\tau^E, h}(s_h, a_h)$ for any state-action pair in the rollout dataset by substituting $s_{h + 1}$ into $s'$, as described in Algorithm 3.
>
>     For non-deterministic environments, we need to assume the realizability of the intrinsic reward function $\mathcal{L}\_{\tau^E}$ (same as Assumption 4.5) and use least-square regression similar as Line 4 of Algorithm 2 to obtain a surrogate $\hat{\mathcal{L}}\_h$ for $\mathcal{L}\_{\tau^E}$: $$\hat{\mathcal{L}}\_h = \arg \min_{\mathcal{L}\_h}\sum_{(s_h, a_h) \in \mathcal{D}\_h} (\mathcal{L}\_h(s_h, a_h) - \\|\hat{s}\_{h + 1}- s_{h + 1}\\|_2^2)^2,$$ where $\hat{s}\_{h+1}$ is the predicted next state after taking $a_h$ at state $s_h$. We've added a remark in our paper to explain how to deal with this situation.
>
> * **Regarding Assumption 4.1:**
>
>     Here, we would like to reiterate that we already generalize the previous assumption considered in [1], where all the reward functions are assumed to be linear functions. And since our focus is on the core challenges of imitation learning, we apply the state-of-the-art theoretical results[2] for the online optimization module. Moreover, the current theoretical guarantee for optimization of neural network needs additional assumptions (e.g. Overparameterization Assumption), which in essential are also simplifications of the geometry of the loss landscape. We believe the optimization of more general functions like NNs is a challenging future work of independent interest.
>
> * **Regarding the assumption that the target task to imitate is aligned with seeking novelty**
>
>     We believe that our experimental result is already a strong evidence of the effectiveness of the intrinsic reward, and the empirical evidence and discussions in previous work are sufficient to make this conjecture. We also emphasized the potential limitation in Section 6 and 7.
>
>     In our theoretical analysis, we did not rely on the alignment of extrinsic reward and ''seeking novelty''. So, it is not an assumption we used in our theoretical results. Instead, we are proposing a principled method to optimize the imitation reward with the regularization of ''seeking novelty'' term to avoid simple mimick of demonstration.
>
>     You mentioned that "seeking novelty typically raises safety concerns". We respectfully disagree, as there is no direct evidence supporting this claim. While we acknowledge the importance of safety in imitation learning and reinforcement learning,  this is beyond the focus of this paper. Importantly, all the baselines we compare against also do not consider safety issues/constraints. Our experiments showed that ILDE remarkably outperforms other baselines over a variety of benchmarks. As an exploration bonus, the effectiveness of curiosity reward has also been proved in previous work (e.g., [4]). Extending our approach to scenarios with safety constraints would indeed be an interesting direction for future research.
>
>     [1] Liu et al., Learning from demonstration: Provably efficient adversarial policy imitation with linear function approximation, ICML 2022.
>
>     [2] Orabona F. A modern introduction to online learning[J]. arXiv preprint arXiv:1912.13213, 2019.
>
>     [3] Burda et al., Large-Scale Study of Curiosity-Driven Learning, ICLR 2019.
>
>     [4] D Pathak et al., Curiosity-driven exploration by self-supervised prediction. ICML 2017.
>
>     ----
>
>     Thank you once again for your insightful feedback. Please let us know if you have any further questions or suggestions. If our rebuttal has addressed your concerns, we would greatly appreciate it if you could reconsider your rating.

---

### Official Review · Reviewer_HCFX · 2024-10-31

**Soundness:** 3
**Presentation:** 4
**Contribution:** 3
**Rating:** 8
**Confidence:** 4

**Summary:**

This paper focuses on imitation learning from limited demonstrations, where the goal is to learn a policy that either imitates the expert directly or learns a policy that is better than the expert. Prior techniques focused on exploration-based approaches for this problem, with the goal of designing useful intrinsic motivation bonuses to incentivize the agent to explore its (large) state space to ultimately learn a better-than-expert policy. This paper proposes the use of two bonuses: an exploration bonus that encourages the agent to visit novel state-action pairs in the environment, and another bonus that incentivizes the agent to deviate from the expert trajectories themselves. The authors propose theoretical justification of their method, achieving a poly(H, T) regret w.r.t. the true expert.

**Strengths:**

I am not the best theoretician by any means, but this paper is very theoretically sound. I think the regret analysis was done well, which is the main theoretical contribution of this work.

Experimental results were quite expansive and good. The authors seemed to ablate the right variables (e.g. what happens if we turn our state-action based exploration bonus off, what happens if we turn our imitation reward off) to analyze exactly what is different about their method compared to a standard baseline (e.g. GIRIL). In terms of domains of interest, the authors also covered a variety of benchmarks across a variety of RL benchmarks, including Atari and MuJoCo tasks.

**Weaknesses:**

It would be interesting to see how just the state-action exploration bonus does as well (e.g. the state entropy bonus stays, but the GIRIL-like intrinsic reward is not used). To me, both of these methods seem to be doing the same thing (one on the state-action space as a whole, and one focusing on the expert trajectories), so it would be good to see an experiment that keeps the state entropy bonus, doesn't use the demonstration-based intrinsic reward, and keeps the standard imitation learning reward, which I understand is different from GIRIL (where their intrinsic reward bakes in both the true demonstration actions and the demonstration-based intrinsic reward). This is pretty minor, but would love to see this myself.

**Questions:**

I don't have any additional questions.

---

> ### Author Response · Authors · 2024-11-24
> **Response to Reviewer HCFX**
>
> Thanks for your positive feedback and insightful comments! We answer your questions point-by-point.
>
> **Q1.** Experimental results were quite expansive and good. The authors seemed to ablate the right variables (e.g. what happens if we turn our state-action based exploration bonus off, what happens if we turn our imitation reward off) to analyze exactly what is different about their method compared to a standard baseline (e.g. GIRIL). In terms of domains of interest, the authors also covered a variety of benchmarks across a variety of RL benchmarks, including Atari and MuJoCo tasks.
>
> **A1.** The authors thank the reviewer for the positive evaluation.
>
> ---
>
> **Q2.** It would be good to see an experiment that keeps the state entropy bonus, doesn't use the demonstration-based intrinsic reward, and keeps the standard imitation learning reward.
>
> **A2.** Thanks for your suggestion. We have summarized the results of ILDE without the demonstration-based intrinsic reward (ILDE w/o $\lambda\mathcal{L}_{\tau_E}$) bellow:
>
> |Atari Games | ILDE w/o $\lambda\mathcal{L}_{\tau_E}$ |
> | -------- | -------- |
> |BeamRider | 3,233$\pm$3,340 |
> |DemonAttack | 143$\pm$120  |
> |BattleZone | 5,412$\pm$1,816 |
> |Qbert | 10,849$\pm$5,018 |
> |Krull | 857$\pm$916  |
> |StarGunner | 764$\pm$455 |
> The results show that ILDE w/o $\lambda\mathcal{L}_{\tau_E}$ performs much worse than ILDE on the six Atari games.

---

> > ### Comment · Reviewer_HCFX · 2024-11-26
> >
> > Thanks for the response, and sorry for the late reply. Good to see that the intrinsic reward is in fact crucial.

---

> > > ### Author Response · Authors · 2024-11-27
> > > **Response to Reviewer HCFX**
> > >
> > > Thank you very much for your strong support and helpful feedback! Your thoughtful comments and feedback have been invaluable in helping us improve the paper.

---

### Official Review · Reviewer_4joU · 2024-11-02

**Soundness:** 2
**Presentation:** 2
**Contribution:** 2
**Rating:** 5
**Confidence:** 2

**Summary:**

This paper addresses the challenge of surpassing expert performance through imitation learning. The authors propose ILDE which modifies the Generative Adversarial Imitation Learning (GAIL) framework by incorporating an intrinsic reward term to encourage exploration. The method is supported by theoretical analysis quantifying the algorithm's regret bounds. Empirical evaluation on 6 Atari games and 4 MuJoCo tasks demonstrates ILDE's improvements over baseline methods.

**Strengths:**

- This paper studies a seemingly interesting problem of learning a policy with imitation learning to be much better than the demonstration.
- The paper offer some theoretical analysis of the proposed method.

**Weaknesses:**

- The problem formulation of optimizing the GAIL loss + intrinsic reward to achieve a better performance than the expert is not convincing.
- The proposed method is only evaluated on limited tasks and the results lack explanation.

**Questions:**

Majors:
- As mentioned in the weakness part, the biggest concern from me is the formulation. Could you please elaborate why, intuitively, adding an intrinsic reward term to the GAIL objective (Eq. 3.3) can help the policy to achieve a better performance than the expert? I think to achieve a better performance than the expert, the agent must understand the intention of the expert and extrapolate it. I cannot see how an intrinsic reward guarantee to help for this. Maybe there are some hidden assumptions about the task or the demonstrations?
- In the introduction, you list the formulation of the problem itself as one of the contribution, but you write the formulation in the Preliminary section. Could you elaborate what is the difference between the formulation in this paper with previous works, for example [1]?
- From the results in Table 1, ILDE w/o $r^k$ is much better than ILDE w/o $b$. Is this a desired behaviour that using only the intrinsic reward allows achieving a better performance than the imitation reward? Does this not imply that the selected 6 Atari tasks can be solved by only doing exploration?

Minors:
- In line 41, your citation of GAIL is wrong.
- No number is bold in the Qbert line of Table 1.
- In the result section, you only include Table 1 in the main text but spend the majority of the text discussing the results on Tables 9 and 10 in the appendix. Can you reorganize the paper to make the text and table better align with each other?

[1] Yu *et al.*, Intrinsic Reward Driven Imitation Learning via Generative Model, ICML 2020

---

> ### Author Response · Authors · 2024-11-23
> **Response to Reviewer 4joU**
>
> Thanks for your helpful comments. We address your questions as follows.
>
> **Q1.** The problem formulation of optimizing the GAIL loss + intrinsic reward to achieve a better performance than the expert is not convincing. Could you please elaborate why, intuitively, adding an intrinsic reward term to the GAIL objective (Eq. 3.3) can help the policy to achieve a better performance than the expert?
>
> **A1.** As shown in [2], by optimizing the classic objective one could not expect to achieve beyond-expert performance even with a large amount of demonstration trajectories. Thus, it is natural to consider self-supervised RL frameworks which do not rely on explicit rewards but make use of the potential correlation between the transition dynamics and reward function.
> In [3], large scale experiments have shown that dynamics-based curiosity is a powerful self-supervised exploration strategy in a variety of environments. In their paper, it is discussed that **empirical RL environments typically require the learner to learn non-trivial skills which coincides with the curiosity-reward signal that encourages the learner to explore hard-to-visit states**. The intrinsic reward proposed by [1] is essentially a variant of the curiosity reward which uses VAE model to better characterize the uncertainty of state-action pairs. Previous works [1, 3] have provided **sufficient** empirical evidence of the effectiveness of the intrinsic reward and our experiments show that this is indeed the case. The main contribution of our paper is to provide a novel and principled way to optimize Eq. 3.3. We note that there is limited theoretical understanding of curiosity reward at this point, and we think this is a challenging but important avenue for future works. For readers to better understand this intuition, we would also include this discussion in the revision.
>
> **Q2.** The proposed method is only evaluated on limited tasks and the results lack explanation.
>
> **A2.** The authors respectfully disagree with this statement as there are already 6 Atari games and 4 MuJoCo games in the paper, which is more than related work (e.g. HyPE’s 6 total number of tasks and GAIL’s 9 total number of tasks).
>
> **Q3.** In the introduction, you list the formulation of the problem itself as one of the contribution, but you write the formulation in the Preliminary section. Could you elaborate what is the difference between the formulation in this paper with previous works, for example [1]?
>
> **A3.** In contrast to [1] which purely relies on the intrinsic reward to explore transitions that are distinct from the expert demonstration, we consider learning a policy which mimics the expert policy subject to an intrinsic reward regularization term.
> Additionally, we are interested in how to solve the objective sample-efficiently, for which we introduce an additional exploration bonus.
>
> **Q4.** From the results in Table 1, ILDE w/o $r_k$ is much better than ILDE w/o $b$. Is this a desired behavior that using only the intrinsic reward allows achieving a better performance than the imitation reward? Does this not imply that the selected 6 Atari tasks can be solved by only doing exploration?
>
> **A4.** As we demonstrate in Table 1, each component of the reward function contributes positively to the performance, demonstrating each component is necessary.
>
> **Q5.** In line 41, your citation of GAIL is wrong.
>
> **A5.** Thank you for pointing this out. The authors have corrected the reference.
>
> **Q6.** In the result section, you only include Table 1 in the main text but spend the majority of the text discussing the results on Tables 9 and 10 in the appendix. Can you reorganize the paper to make the text and table better align with each other?
>
> **A6.** Thank you for your suggestion. We have updated the paper to reorganize the result description in Section 6.1.
>
> [1] Yu et al., Intrinsic Reward Driven Imitation Learning via Generative Model, ICML 2020.
>
> [2] Nived Rajaraman et al., Toward the fundamental limits of imitation learning, NIPS 2020.
>
> [3] Burda et al., Large-Scale Study of Curiosity-Driven Learning, ICLR 2019.

---

> > ### Comment · Reviewer_4joU · 2024-11-25
> >
> > I would like to thank the authors for their rebuttal and clarification. I am sorry if it takes a bit long for me to reply. After carefully reading all reviewers' comments, the authors' rebuttals and revisiting some sections of the paper, the method does make more sense. I was misunderstood that the intrinsic reward is independent with the demonstrations, but seems like $L_{\tau_E}$ has the dependency.
> >
> > However, I am still concerned that the method seems to be only evaluated on the environments where the intrinsic motivation is aligned with the task. In the discussion of Burda *et al.* [3], they said "More generally, these results suggest that, in environments designed by humans, the extrinsic reward is perhaps often aligned with the objective of seeking novelty. The game designers set up curriculums to guide users while playing the game, explaining the reason Curiosity-like objective decently aligns with the extrinsic reward in many human-designed games." This discussion implies that the game environments has a strong bias of the alignment between the intrinsic motivation and the human-designed reward. This is also supported by your experiments. In your new Table 1, apparently among the three rewards, the intrinsic one $L_{\tau_E}$ plays the most important rule while the imitation one $r_k$ plays the least important rule. Then on the MuJoCo environments, the performance boost is marginal comparing with the imitation-only algorithm VAIL. And it could also be explained by the survival intend enforced by the intrinsic reward, i.e. the policy need to survive as long as possible to collect the intrinsic reward without trigger the terminal condition of the environments. All these evidence makes me question about the applicability of the method to complicate real problems that are **not** carefully designed by human to be game-like.
> >
> > To make my argument concrete, let's do a thought experiments. Say you are learning a policy for a humanoid, and you are getting some demonstrations that move around 1 m/s forward with some weird style, e.g. jumping by one leg. But then the task is undefined, right? The task can be to move forward as fast as possible, or to maintain a stable movement with only one leg as long as possible, or even to maintain a constant speed of 1 m/s. For which task will your intrinsic motivation help you toward to?
> >
> > Currently, the paper has no mentions about these limitations.

---

> > > ### Author Response · Authors · 2024-11-26
> > > **Response to Reviewer 4joU**
> > >
> > > Thank you very much for reading our rebuttal and for your feedback! We are glad that our response has clarified your previous misunderstanding of the intrinsic reward. Here we would like to address your remaining concerns as follows.
> > >
> > > Regarding the experiments on Atari and MuJoCo, it is worth mentioning that the intrinsic reward itself does not enforce survival intent because we keep normalizing the reward to zero mean value (same trick has also been used in [1]) to stabilize the training process in our implementation. Hence, we suspect that your explanation that the intrinsic reward encourages the policy to survive as long as possible may not be true.
> > >
> > > Here, we reiterate the excerpts from [1] that were referenced in your feedback.
> > > > "More generally, these results suggest that, in environments designed by humans, the extrinsic reward is perhaps often aligned with the objective of seeking novelty. The game designers set up curriculums to guide users while playing the game, explaining the reason Curiosity-like objective decently aligns with the extrinsic reward in many human-designed games." from [1]
> > >
> > > We believe the most reasonable explanation for the intrinsic reward is that it is generally **aligned with the goal of seeking novelty**, which is closely tied to exploration in reinforcement learning. This serves as the motivation and foundation for our work, where we aim to encourage exploration through intrinsic rewards in imitation learning, achieving beyond-expert performance.
> > >
> > > We also agree that the alignment between intrinsic motivation and the human-designed reward is a key factor in the effectiveness of intrinsic rewards. However, this is just one example of seeking novelty driven by curiosity.
> > >
> > > That being said, we appreciate your suggestion to include a discussion on the limitations of our work. In response, we have added the following sentence: "Based on our current experiments, we observed that ILDE performs exceptionally well in games, while its improvement on MuJoCo is less significant. This may suggest that the current design of the intrinsic reward is better suited for tasks with human-designed rewards."
> > >
> > > Thank you once again for your insightful feedback. Please let us know if you have any further questions or suggestions. If our rebuttal has addressed your concerns, we would greatly appreciate it if you could reconsider your rating.
> > >
> > > [1] Burda et al., Large-Scale Study of Curiosity-Driven Learning, ICLR 2019.

---

> > > > ### Comment · Reviewer_4joU · 2024-11-26
> > > >
> > > > Thanks for your further clarification. I am not sure whether it is a typo in your reply. But we have to be clear here:
> > > > - "intrinsic motivation" means the same as "seeking novelty" in the context of this paper.
> > > > - The reason why this works well on the six Atari games is, as the quote says, for these environments the **extrinsic reward** is by design aligned with "seeking novelty".
> > > >
> > > > If we can agree on these, I strongly suggest making this clear in the paper that the method assumes the target task to imitate is aligned with seeking novelty, or curiosity. It can achieve performance beyond the demonstration is not because the algorithm can understand and extrapolate the intention in the partial demonstration.
> > > >
> > > > I appreciate the new sentence about the limitation, but I think the word "human-designed" is too vague. Essentially, every reward we use, also for the MuJoCo tasks, is human-designed. I would more emphasize about its alignment with the intrinsic motivation or seeking novelty.

---

> > > > > ### Author Response · Authors · 2024-11-27
> > > > > **Response to Reviewer 4joU**
> > > > >
> > > > > Thank you for your further feedback on our rebuttal and revision!
> > > > >
> > > > > Yes, we agree. To put it another way, the reason our algorithm can achieve beyond-expert performance is that, in these environments used in our experiments, the extrinsic reward is inherently aligned with the objective of "seeking novelty", and our intrinsic reward is designed to promote novelty-seeking, effectively serving as a surrogate for the extrinsic reward.
> > > > >
> > > > > Following your suggestion, we have revised our discussion in Section 6.2 and added a paragraph called "Limitation of our work" in the conclusion section as follows:
> > > > >
> > > > >     "Limitation of our work. Our method assumes that the target task to be imitated is aligned with the concept of seeking novelty or curiosity. Its ability to achieve performance beyond the demonstration is not due to the algorithm's capacity to understand and extrapolate the intention behind the partial demonstration. "
> > > > >
> > > > > Please let us know if you have any further feedback and suggestions. Thank you.

---

> > > > > > ### Comment · Reviewer_4joU · 2024-11-27
> > > > > >
> > > > > > The revision looks good. I will raise my score to 5 since as we clarified the method may not be applicable for many real-world tasks due to the strong assumption.
> > > > > >
> > > > > > In the meantime, the paper seems to make a solid contribution on the theory part, but I am not the right person to judge that. I will depend on Reviewer FEJp who raise the similar concerns as me to judge that part.
> > > > > >
> > > > > > I may still change my score based on the final outcome of the discussion.

---

> ### Author Response · Authors · 2024-11-27
> **Thank you!**
>
> Thank you for increasing the score. We are delighted that our rebuttal and revision have addressed your major concerns.
>
> While much of our discussion has focused on formulation, algorithm design and its underlying rationale, and experiments, we want to emphasize that theoretical analysis is also a key contribution of our work.
>
> Reviewer FEJp expressed concerns that Assumptions 4.1 and 4.2 in our theoretical analysis might be too strong. As we clarified in our rebuttal to Reviewer FEJp, Assumption 4.2 follows directly from the Azuma-Hoeffding inequality, and Assumption 4.1 is a standard assumption in optimization. Moreover, as discussed in our rebuttal, Assumption 4.2 is actually more general than its counterpart in prior analyses of online imitation learning. We hope this explanation also helps address any related concerns you might have.
>
> Thank you once again for your thoughtful comments and helpful feedback, which have been invaluable in improving our paper.

---

### Meta-Review · Area_Chair_2nCV · 2024-12-19

**Metareview:**

The paper talks about imitation learning with an additional exploration bonus and claims that it can learn a policy that outperforms the expert. The strength of the paper is that it empirically demonstrates that adding both types of bonus is actually quite effective for the imitation learning algorithm.

The weakness comes from the application scope of the proposed approach, particularly under what situations using an exploration bonus can allow the IL agent to perform better than the expert. We encourage the authors to explicitly address this concern in the revised version since, in general, without any additional information (e.g., ground truth reward), it is hopeless for IL to outperform the expert, and an exploration bonus can, in worst case guide the learner to a low-reward region instead of the high-reward region.

**Additional Comments On Reviewer Discussion:**

The authors addressed the concerns from one negative review by explaining their assumption is standard (just concentration), and their algorithm does not require a known transition, and adding additional experiments with HyPE as the baseline algorithm. The author also addressed a concern from another negative review regarding when we should expect the exploration bonus to help the imitation learner.

---

### Decision · Program_Chairs · 2025-01-22

Accept (Poster)